# Barents-2.5km v2.0: An operational data-assimilative coupled ocean and sea ice ensemble prediction model for the Barents Sea and Svalbard

Johannes Röhrs[1], Yvonne Gusdal[1], Edel Rikardsen[1], Marina Duran Moro[1], Jostein Brændshøi[2], Nils Melsom Kristensen[1], Sindre Fritzner[3], Keguang Wang[1], Ann Kristin Sperrevik[1], Martina Idžanović[1], Thomas Lavergne[1], Jens Debernard[1], and Kai H. Christensen[1,4]

[1]Norwegian Meteorological Institute, Henrik Mohns Plass 1, 0371 Oslo, Norway
[2]Norwegian Defence Research Establishment, Instituttveien 20, 2007 Kjeller, Norway
[3]UiT The Arctic University of Norway, Postboks 6050 Langnes, 9037 Tromsø, Norway
[4]Department of Geosciences, University of Oslo, PO Box 1022, Blindern, NO-0315 Oslo, Norway

**Correspondence:** J. Röhrs (johannes.rohrs@met.no)

**Abstract.** An operational ocean and sea ice forecast model, *Barents-2.5*, is implemented for short-term forecasting at the coast off Northern Norway, the Barents Sea, and the waters around Svalbard. Primary forecast parameters are sea ice concentration (SIC), sea surface temperature (SST), and ocean currents. The model also provides input data for drift modeling of pollutants, ice berg, and in search-and-rescue applications in the Arctic domain. *Barents-2.5* has recently been upgraded to include an Ensemble Prediction System with 24 daily realizations of the model state. SIC, SST, and *in-situ* hydrography are constrained through the Ensemble Kalman Filter (EnKF) data assimilation scheme executed in daily forecast cycles with lead time up to 66 hours. Here, we present the model setup and validation in terms of SIC, SST, *in-situ* hydrography, as well as ocean and ice velocities. The performance of the ensemble to represent the model's uncertainty, and the performance of the EnKF to constrain the model state are discussed, in addition to the model's forecast capabilities for SIC and SST.

## 1 Introduction

Rapid changes in water mass distribution and seasonal sea ice cover are underway in the Barents Sea and the areas around Svalbard (Lind et al., 2018). We also see changes in human activities such as shipping, oil exploration, and fisheries. The ability to provide efficient emergency services in this region for search-and-rescue, accidental oil spills, ship drift, etc. is much reduced compared to mainland Norway due to the vast distances and challenging environmental conditions. Contingency models for decision support are based on short-term forecast models for weather, surface waves, ocean circulation and sea ice distribution, and increased predictability in these models translates directly into better emergency preparedness (Röhrs et al., 2023). Of particular interest are upper-ocean transport processes (e.g. Strand et al., 2017, 2021) and energy exchanges across the air-ice-sea interfaces.

In addition, Arctic Ocean ecosystems are vulnerable to change. Projected long term decline in sea ice cover, and the associated "Atlantification" of the Barents Sea (Asbjørnsen et al., 2020), is impacting key species at different trophic levels and hence

an important topic for investigation (Ingvaldsen et al., 2021). Hence, accurate models of upper-ocean transport processes are also important for understanding connectivity of fish stocks and other physical-biological couplings (Röhrs et al., 2014; Strand et al., 2017).

A high-resolution (2.5 km) numerical weather prediction (NWP) system at the Norwegian Meteorological Institute (MET), Arome-Arctic (MET-AA), for the Barents Sea and the areas around Svalbard was established in 2015 (Müller et al., 2017). This development paved the way for a coupled ocean and sea ice forecast model that operates on the same horizontal grid. Here we present this coupled ocean – sea ice forecasting system, the *Barents-2.5* model.

Ocean forecasting in this region is challenging for several reasons: The first challenge is that the hydrography is poorly observed and *in-situ* data are scarce compared to other regions. We have frequent passes of polar orbiting satellites, but since the skies seldom are clear, we have relatively few high-resolution observations of sea surface temperature (SST) from infrared sensors. We do have abundant medium-resolution observations of the sea ice cover from passive microwave imagery (Spreen et al., 2008), and frequent high-resolution images from synthetic aperture radars (SAR). Combined, these sensor technologies provide sub-daily coverage in the region.

A second challenge in high latitudes is that the atmospheric low pressure systems can be small but very intense (Noer et al., 2011; Furevik et al., 2015). Correctly describing the temporal and spatial development of polar lows in the NWP system is difficult, although progress is being made with improvements in physics (Batrak and Müller, 2019) and in the data assimilation (DA) methodology (Hallerstig et al., 2021; Mile et al., 2022). We have, however, significant knowledge gaps when it comes to rapid changes in upper ocean conditions and sea ice cover due to complex air-sea interactions including waves. In addition, many surface layer processes are quite crudely parameterised in the models, if at all.

A third challenge is that the inertial period and the dominant tidal period (M2) are very similar at the latitudes considered here (e.g. Röhrs and Christensen, 2015). Hence, it is difficult to disentangle the transient Ekman dynamics from the more predictable tidal circulation. A general lack of long observation time series of ocean currents or mean sea level implies that we cannot easily remove the tidal component of observed drift velocities to isolate effects of air-sea interactions on the upper ocean transport.

Ocean forecasting in high latitudes must encompass the uncertainties that result from the above challenges. As widely applied in NWP, Ensemble Prediction Systems (EPS) can directly quantify the uncertainties in a forecast. Ensemble prediction has not yet been widely used in regional ocean modeling, but recent works have shown that models may predict uncertainty correctly if the underlying ocean model exhibits statistical skill (Jacobs et al., 2021). Our goal has been to implement an operational EPS for the Barents Sea that assimilates the available observations in this region. The uncertainty provided by the EPS may be used directly in forecast applications, e.g. to quantify the reliability of a prediction for decision support. In addition, the ensemble spread can be used to estimate model background errors, which is needed in DA schemes such as the Ensemble Kalman Filter (EnKF) applied in this work (Evensen, 1994; Sakov and Oke, 2008a).

This paper describes the operational setup of the *Barents-2.5* forecast model. In section 2, we provide details on the configuration and setup of the ocean and sea ice model, including the coupling scheme and forcing data. Section 3 describes the DA scheme and the assimilated observations. All observation data used for DA and for validation in this paper are described in

section 4. Section 5 documents how the model is operationalized, which includes details on the setup as an EPS. In section 6 we describe methods used for validation of the model. Results on model validation and performance are shown in section 7, focusing on validation of SST and sea ice concentration (SIC) as central state variables. Finally, in section 8, we discuss the model's capability as forecast tool and inform about our plans for future model development.

## 2 Model physics and configuration

### 2.1 Ocean circulation model

*Barents-2.5* is built on the Regional Ocean Modeling System (ROMS) version 3.7, which applies a topography-following coordinate system in the vertical (Shchepetkin and McWilliams, 2005) and a curvilinear horizontal grid. ROMS solves the Boussinesq primitive equations. The modeled state variables are temperature, salinity, surface elevation, and horizontal current velocities. The setup in *Barents-2.5* includes a second-order turbulence closure scheme with turbulent kinetic energy and a generic length scale as state variables (Warner et al., 2005).

The model domain and bathymetry are shown in Fig. 1. The model resolution is approximately 2.5 km, which varies slightly throughout the domain consisting of $737 \times 947$ grid points. The bathymetry is based on the GEBCO global data set and interpolated onto the model grid. Minimum depth is set to 10 m. The bathymetry is smoothed to reduce pressure gradient errors as required in ROMS. Additionally, the coastline has been modified to match numeric grid point criteria by ROMS (i.e., every water grid point must have at least two adjacent open boundaries).

#### 2.1.1 Discretization and advection schemes

The applied vertical discretization uses a stretched, topography-following coordinate system. *Barents-2.5* has 42 layers with higher resolution towards the surface. The stretching of vertical coordinates is configured using the transform function 2 and the stretching function 4 in ROMS, with specific parameters of $\theta_S = 6.0$, $\theta_B = 0.3$, and $H_c = 100$ m. This choice of values results in an upper layer thickness between 0.2 and 1.2 m and maintains an increased resolution in the upper 100 m.

ROMS uses split explicit time stepping, i.e. the barotropic modes are solved using a shorter time step than the baroclinic modes. *Barents-2.5* uses 90 seconds outer time steps for the baroclinic mode to solve the 3D momentum equations, and 30 inner time steps (i.e., 3 seconds) for the solution of 2D momentum, including tides.

Momentum and tracers are advected using a third order upwind scheme in the horizontal and a fourth order centered scheme in the vertical. Turbulent kinetic energy and length scale are advected vertically and horizontally using a fourth order centered scheme.

#### 2.1.2 Sub-scale processes

Vertical mixing is modeled using a second order scheme for turbulent kinetic energy (TKE) and a generic length scale (GLS). Our setup is the one recommended by Umlauf and Burchard (2005). Parameters for this turbulence scheme are documented in

Tab. 1. It is noted that turbulence dissipation rate and turbulent length scale may be calculated from TKE and GLS according to Eqs. 14 and 15 in Warner et al. (2005). The `CANUTO_A` stability function is chosen for the diffusion of momentum and tracers (Canuto et al., 2001; Warner et al., 2005).

The boundary condition for TKE at the surface is based on the model of Craig and Banner (1994) using a flux condition, wherein the energy flux at the surface is proportional to the cube of the air-side friction velocity with a scaling factor of 100. Surface roughness is set by the wind stress using a Charnok constant of 1400. Buoyancy and shear are horizontally smoothed using the `N2S2_HORAVG` pre-compiler option. The background vertical diffusivity is set to $10^{-6}$ m$^2$/s for tracers, $10^{-5}$ m$^2$/s for momentum, and $5 \cdot 10^{-6}$ m$^2$/s for TKE and GLS. Furthermore, the pre-compiler options `RI_SPLINES`, `SPLINES_VVISC`, and `SPLINES_VDIFF` are used.

Harmonic horizontal diffusion of tracers is applied using a diffusivity of 2 m$^2$/s. Explicit harmonic horizontal diffusion is applied using a viscosity of 50 m$^2$/s only in the sponge zone of 30 grid points, increasing from zero to a viscosity of 100 m$^2$/s using an arctangent shaped smooth transition. A harmonic horizontal diffusivity for TKE and GLS is set to 0.1 m$^2$/s

Tracers are mixed along surfaces of constant geopotential, while momentum is mixed along the bottom topography following coordinate surfaces. Quadratic bottom friction is applied using a drag coefficient of 0.003 where the water depth is greater than 100 m. In shallower regions, the bottom drag coefficient is increased up to 0.009 for the shallowest parts with a water depth of 10 m with linear transition as a function of water depth. The bottom drag is limited such that the current cannot reverse sign using the pre-compiler option `LIMIT_BSTRESS` in ROMS. Limiting the bottom drag in such fashion is useful to avoid numeric instabilities in very shallow waters during strong storm surges. A full list of all applied pre-compiler options for ROMS, static files such as grid files, and run-time options are provided in a code repository for *Barents-2.5*[1].

## 2.2 Sea ice model

The sea ice model used in *Barents-2.5* is the Los Alamos sea ice model (CICE) version 5.1 (Hunke et al., 2017). CICE describes both dynamic and thermodynamic processes. In this work, CICE is configured to use elastic-viscous-plastic (EVP) rheology (Hunke and Dukowicz, 1997), which is a compromise for computational cost compared to the elastic-anisotropic rheology (e.g. Heorton et al., 2018). The model solves the evolution of the sea ice state using five ice thickness classes, described by the ice thickness distribution (ITD) function (Thorndike et al., 1975; Hibler III, 1980):

$$\frac{\partial g}{\partial t} = -\nabla \cdot (g\mathbf{u}) - \frac{\partial (fg)}{\partial h} + \psi, \tag{1}$$

where $g(\mathbf{x}, h, t) \, dh$ is defined as the fractional area covered by ice in the thickness range $(h, h + dh)$ at a given time $t$ and location $\mathbf{x} = (x, y)$, $f$ is the rate of thermodynamic ice growth, and $\psi$ is a ridging redistribution function. The ice velocity $\mathbf{u}$ is calculated from the sea ice momentum equation that accounts for air and water drag, Coriolis force, sea surface tilt, and the divergence of internal ice stress. The evolution of internal stress is described by the EVP rheology, with the ice strength reformulated according to Rothrock (1975), and the advection using the incremental remapping scheme (Lipscomb and Hunke, 2004). The

---

[1]https://doi.org/10.5281/zenodo.7607191

subgrid sea ice deformation and the redistribution of various ice categories follow (Rothrock, 1975), with a modified expression for the participation function (Lipscomb et al., 2007).

The sea ice thermodynamic growth and melting $f$ is determined by solving the one-dimensional vertical heat balance equations for each ice thickness category and snow. The sea ice heat balance equation is solved with the mushy-layer scheme that also accounts for the evolution of sea ice salinity (Turner et al., 2013). The upper snow and ice boundary is assumed to be in balance with short- and long-wave radiation and the sensible, latent and conductive heat fluxes when the surface is below freezing point. When the surface temperature reaches the melting point, it is held constant and the extra heat is used to melt the snow and ice surface. The bottom water-ice interface is assumed to be at thermodynamic balance, such that growing or melting results from the difference between ice conductive heat flux and the under-ice oceanic heat flux. An example of the CICE input parameter file, including all choices of numeric schemes and input parameters is provided in the code repository for *Barents-2.5* [2].

## 2.3 Ocean-ice coupling

The ROMS-CICE coupling utilizes the Model Coupling Toolkit (MCT, Larson et al., 2005) for inter-model exchange of state variables and fluxes, as implemented in the *METROMS* framework [3] (Debernard et al., 2021). The coupling was briefly explained in Naughten et al. (2017) and Naughten et al. (2018), and in more detail by Duarte et al. (2022). The underlying philosophy behind the coupling is that surface fluxes are calculated in the component with most information about the surface, utilizing required information from the other component. Finally, the heat, water, and momentum fluxes are passed back to the other component. In *METROMS*, the coupling is based on the principle of 'levitated' ice, so there is no actual exchange of mass between the ocean and the ice. Freshwater and salt fluxes from the ice model are converted to a virtual salt flux before they are used in ocean model. In this 'massless' state, the ice does not displace water, and it is only seen by the ocean as a source of surface fluxes responding to the present ocean state. The ice and ocean models are run concurrently, with exchange of information every baroclinic ocean time step. The information used by each model is therefore time-lagged compared with its own state.

## 2.4 Boundary conditions and river input

The boundary conditions for *Barents-2.5* are provided by *Topaz* version 4, a coupled ocean and sea ice DA system based on the Hybrid Coordinate Ocean Model (HYCOM) ocean model and an EnKF DA scheme. *Topaz4* is configured for the North Atlantic and Arctic Ocean with a horizontal resolution of 12-16 km (Xie et al., 2017). It provides daily averages of temperature, salinity, sea surface elevation, and ocean current velocities for the ocean component. For the sea ice component, it provides daily averages of SIC, sea ice thickness, first year ice age, snow depth, and ice velocity. The inverse barometric effect due to the local atmospheric pressure is added to the sea surface elevation boundary conditions because *Topaz4* does not include the barotropic signal of atmospheric pressure.

---

[2]https://doi.org/10.5281/zenodo.7607191

[3]https://github.com/metno/metroms.git

The numeric schemes for boundary conditions used in ROMS are given in Tab. 2, and they differ for the various state variables. A sponge zone with up to 10-fold increased horizontal tracer diffusivities and viscosity is implemented within 40 grid points from the boundary. Nudging of passive tracers towards the boundary fields from *Topaz4* is imposed within the sponge zone. 2D momentum anomalies are radiated out of the model domain using the tangential phase speed of the barotropic signal, using ROMS' `RADIATION_2D` option. Details about the implementation of boundary conditions in CICE are given in Duarte et al. (2022).

Point sources for river influx along the coast in the model domain are shown in Fig. 1. At each of the 318 river locations, daily values for temperature, salinity, and mass flux are specified. Climatological values for rivers in the Svalbard archipelago are used, while river data on mainland Norway originates from daily estimates provided by the Norwegian Water Resources and Energy Directorate.

Tidal forcing is provided as amplitudes and phases of the ten major tidal constituents in the model domain (Tab. 3). These are obtained from the TPXO global inverse barotropic model (Egbert and Erofeeva, 2002) and imposed on velocities and free surface elevation. The tidal signal is also added to velocities and surface elevation during the processing of boundary data.

## 2.5 Atmospheric forcing

Surface forcing in *Barents-2.5* is provided by ensemble forecasts from the Integrated Forecast System (IFS) at European Centre for Medium Weather Forecasts, hereafter referred to as ECMWF-ENS, which has a horizontal resolution of about 18 km. Hourly surface fields of wind speed, air temperature, humidity, rain fall, and cloud cover are used by the bulk flux module of ROMS to calculate surface fluxes as upper boundary condition in the ocean. CICE uses air temperature, humidity, density, precipitation rate and winds to calculate surface stress, heat fluxes and snow aggregation on sea ice.

*Barents-2.5* is configured as an EPS with 24 members, wherein 20 members are forced with random members from ECMWF-ENS and four members are forced by MET-AA (Müller et al., 2017). MET-AA is nested into the IFS high-resolution forecasts.

## 3 Data assimilation scheme

The model state in *Barents-2.5* is constrained by an EnKF DA scheme (Evensen, 1994; Burgers et al., 1998; Houtekamer and Mitchell, 1998; Evensen, 2003). Central in this scheme are the model error, estimated by the spread of the EPS, and the observation errors. A new analysis is obtained on a daily basis, which is aimed to reduce the errors in the model state compared to the available observations. The assimilated variables are SST, SIC, and *in-situ* hydrography, i.e. salinity and temperature observations (Tab. 4).

## 3.1 The deterministic Ensemble Kalman Filter

The EnKF is a sequential ensemble based assimilation method that has been used for many geophysical applications (Houtekamer and Zhang, 2016). The standard analysis equation solved by the EnKF is given by

$$\mathbf{x}_a = \mathbf{x}_b + \mathbf{P}_b \mathbf{H}^T \left( \mathbf{H} \mathbf{P}_b \mathbf{H}^T + \mathbf{R} \right)^{-1} (\mathbf{y} - \mathbf{H}\mathbf{x}_b), \tag{2}$$

where $\mathbf{x}_a \in \mathbb{R}^{n \times N}$ is the analysis state vector representing the updated variables after assimilation, $\mathbf{x}_b \in \mathbb{R}^{n \times N}$ is the model background state, and $\mathbf{y} \in \mathbb{R}^{m \times N}$ is the observation vector. $N$ is the number of ensemble members, $n$ is the number of variables multiplied by the number of spatial grid points in our model, $m$ is the total number of observations, $\mathbf{R} \in \mathbb{R}^{m \times m}$ is the observation covariance matrix, and $\mathbf{H} \in \mathbb{R}^{m \times n}$ is the observation operator. The Kalman gain $\mathbf{K}$ defined as $\mathbf{P}_b \mathbf{H}^T \left( \mathbf{H} \mathbf{P}_b \mathbf{H}^T + \mathbf{R} \right)^{-1}$ in Eq. 2 corresponds to the weigh given to the observations and current background state. The key property of the EnKF is that the background error covariance matrix $\mathbf{P_b} \in \mathbb{R}^{n \times n}$, approximating the model uncertainty and covariance between variables and locations, is estimated as the flow-dependent covariance of the ensemble of background states as

$$\mathbf{P_b} = \overline{((\mathbf{x}_b - \overline{\mathbf{x}_b})(\mathbf{x}_b - \overline{\mathbf{x}_b})^T)}. \tag{3}$$

In the equation above, the overbars denote the ensemble average operator.

In this study, the deterministic version of the EnKF (Sakov and Oke, 2008a) is applied. The same DA scheme was also applied in a preceding study with *Barents-2.5*, assimilating only sea ice variables (Fritzner et al., 2019, 2020). The Deterministic Ensemble Kalman Filter (DEnKF) does not require perturbation of observations to maintain ensemble spread as is required by the original formulation of the EnKF. For the stochastic EnKF, perturbations are added to the ensemble equivalence of the observation, accounting for non-symmetric observation errors (van Leeuwen, 2020). Perturbing observations introduces additional sampling error in the analysis, which for applications with few ensemble members might be a significant contribution (Sakov and Oke, 2008a; Whitaker and Hamill, 2002). In the DEnKF, the assumption of a small $\mathbf{KH}$ term leads to a simplified ensemble transform matrix (ETM) used to update the ensemble anomalies. The left ETM $\mathbf{T_L} = (\mathbf{I} - \mathbf{KH})^{1/2}$ (Sakov and Oke, 2008b) is expanded into a Taylor series with only the two first terms being retained, resulting in $\mathbf{A_a} = (\mathbf{I} - 1/2\mathbf{KH})\mathbf{A_f}$, where $\mathbf{A_a}$ and $\mathbf{A_f}$ are the analysis and forecast ensemble anomalies, respectively. The DEnKF analysis scheme consists of four main steps. First, the forecast ensemble mean is computed as $\mathbf{x_f} = 1/m \sum_{i=1}^{m} \mathbf{X_f^i}$ and the forecast ensemble anomalies as $\mathbf{A_f^i} = \mathbf{X_f^i} - \mathbf{x_f}$, with $i = 1..m$ where $m$ is the ensemble size and $\mathbf{X_f^i}$ each ensemble model state. Then, the analysis $\mathbf{x_a}$ is computed using Eq. 2. The analysed ensemble anomalies are computed using the ETM $T_L$ previously defined. Finally, the analysed ensemble is provided by $\mathbf{X_a} = \mathbf{A_a} + [\mathbf{x_a}, ..., \mathbf{x_a}]$. More details on the theoretical and technical aspects of the DEnKF can be found in Sakov and Oke (2008a) as well as in the *EnKF-C* user manual[4].

## 3.2 Implementation of the data assimilation method

The implementation of the DEnKF in *Barents-2.5* is accomplished using the *EnKF-C* software package. The DEnKF is executed offline, i.e., in between daily forecast cycles, thereby providing updated initial conditions $\mathbf{x_a}$ to ROMS and CICE at 00 UTC each day. Observations $\mathbf{y}$ taken during the previous 24 hours are synchronously evaluated and compared with the model state $\mathbf{x_b}$ at time 00 UTC, hence assuming that observed variables retain validity for 24 hours. The limits of these assumption are discussed in section 8.2.

---

[4]Sakov, P.: EnKF-C user guide, https://doi.org/10.48550/arXiv.1410.1233, 2014

*In-situ* temperature and salinity are directly mapped to the respective model state variables. SST observations are compared to the upper layer model temperature. We note that surface skin temperature may differ from the upper layer temperature and comment on this simplification in section 8.2.

While CICE models the sea ice cover using five ice thickness classes with SIC for each class, observations of SIC are compared with aggregated SIC from all classes, wherein we ensure that the analysis increments do not exceed the lower and upper bounds of aggregated SIC, e.g. $0 < \text{SIC} < 1$, by capping the model increments after the EnKF step. While the EnKF formerly requires Gaussian distribution of variables across the ensemble, SIC distributions deviate from Gaussian distribution due to the lower and upper bounds. In section 8.2, the ensemble distribution of SIC at various model locations is evaluated.

### 3.3 Configuration of *EnKF-C*

The *EnKF-C* software package allows configuration of the DA scheme in terms of an exaggeration parameter $R$, a localisation radius $r_{loc}$ for each observation type (Tab. 4), a global moderation factor $K = 1.5$, and an inflation factor. The latter applies a scaling of the model increments; when set above 1 the ensemble anomalies are inflated. This inflation factor is set to 1.05 (5% inflation) for all model variables except for SIC using 1.1 (10% inflation); a higher inflation factor is required for SIC compared to other variables because of its lower ensemble spread, which is particularly needed around the ice edge area.

The impact of observations is moderated through the observation errors, the exaggeration factor $R$, and the moderation factor $K$. The used *in-situ*, SST, and SIC observations provide observation errors along with observed values. $R$ is tuned to balance the effectiveness of analysis increments while maintaining sufficient ensemble spread in the model. $K$ regulates the moderation of the observation impact by smoothly increasing the observation error as a function of the innovation magnitude. For large innovations, $K$ plays an important role in the increase of the observation error; for small innovations, observations errors are mostly kept unchanged. For details on the observation error moderation we refer to Sakov and Oke (2008a) and the user manual for *EnKF-C* [5].

The localisation radius $r_{loc}$ in *EnKF-C* allows to perform the DA analysis based on a horizontal subset of the full model domain, i.e. limiting the covariance matrices to non-zero entries only within the localisation area. In essence, this allows us to use the EnKF method for a system with large degrees of freedom with a relatively low number of ensemble members. The *EnKF-C* software does only provide a horizontal localisation, hence no vertical localisation is applied.

## 4 Observations

Observations used for DA in *Barents-2.5* are *in-situ* temperature and salinity, SST from satellite, and passive microwave SIC. For validation, the same observations are used in addition to subjective ice charts, passive microwave ice drift vectors, and radial currents from a high-frequency (HF) radar station at the coast of Northern Norway. A subset of *in-situ* temperature and salinity data that have not been assimilated are presented separately during the validation of the model results. Access paths for all used data are given in the code and data availability section.

---

[5] Sakov, P.: EnKF-C user guide, https://doi.org/10.48550/arXiv.1410.1233, 2014.

## 4.1 *In-situ* temperature and salinity

*In-situ* observations of salinity and temperature are routinely collected from the *in-situ* thematic assembly centre of the Copernicus Marine Environment Monitoring Service (CMEMS)[6]. The data set consists of observations from a variety of platforms, such as drifting buoys, profiling floats, moorings, Conductivity-Temperature-Depth (CTD) casts, and thermosalinographic data from monitoring cruises and the Ships-of-Opportunity Program. As an illustration of the quantity of this *in-situ* data within the *Barents-2.5* domain, Fig. 2 shows the spatial location of this data for a specific date in 2022. Additional *in-situ* observations are retrieved from the Global Telecommunication System (GTS) and from MET's observation database[7]. The observations are run through basic quality control procedures prior to DA, and duplicate observations between different sources are removed.

## 4.2 Sea surface temperature

Level 2P satellite SST from the infrared sensors AVHRR (Metop satellites), VIIRS (Suomi NPP and NOAA-20 satellites), and SLSTR (Sentinel-3 mission) are used for DA. A bias correction scheme is applied to ensure consistency between the different SST products. The SLSTR measurements are used as a reference because this instrument uses a dual-view technique which gives more accurate SST measurements. The bias correction scheme is described in more detail in Iversen et al. (2023).

All *in-situ* and SST observations are processed using the python package pyromsobs[8]. When more than one observation of a given state variable is available within the same grid cell at the same time, they are replaced with a so-called super-observation, which is a mean of the available observations.

## 4.3 Sea ice concentration

Observations of SIC are computed from microwave brightness temperature observations from the Advanced Microwave Scanning Radiometer-2 (AMSR2) sensor on board of the Global Change Observation Mission – Water "Shizuku" (GCOM-W1) satellite. The SIC algorithm *SIRANO*[9] is an evolution of the algorithm described in Lavergne et al. (2019), which involves dynamic tuning of the algorithm coefficients and atmospheric correction of the brightness temperatures using a radiative transfer model. Two SIC fields are computed independently and then combined before they are assimilated: first a low resolution, low uncertainty SIC from the 18.7 and 36.5 GHz imagery channels, then a higher resolution, higher uncertainty SIC from the 89.0 GHz imagery channels. The two SICs are then combined using a pan sharpening method to preserve the higher resolution and low noise characteristics from each. Because the nominal spatial resolution of the 89.0 GHz imagery of AMSR2 is 3×5 km, the resulting SIC fields have a spatial resolution slightly coarser than the 2.5 km grid spacing of the model.

In the operational setup, SICs are computed for each incoming satellite orbit (every 100 minutes) and then combined and projected to cover the geographic domain of the model in 12-hourly fields, including uncertainties. Because of the orbit of

---

[6]http://marine.copernicus.eu

[7]http://frost.met.no

[8]https://github.com/metno/pyromsobs

[9]Rusin, J., T. Lavergne, A. P. Doulgeries, K. A. Scott. Resolution enhanced sea ice concentration: a new algorithm applied to AMSR2 microwave radiometry data. Submitted to Annals of Glaciology.

the GCOM-W1 satellite, the domain is observed in the period from 10 UTC of the previous day to 02 UTC in the morning. Data from this period are used for an assimilation at 00 UTC model time. The observed SIC is compared to the model SIC accumulated over all thickness classes.

## 4.4 Subjective ice charts

The ice charts are based on a manual production by ice analysts at MET who define the ice edge and ice classes. Subjective ice charts are produced daily from Monday to Friday and provided with a 1 km horizontal resolution, based on Sentinel-1A/B synthetic aperture radar (SAR) images taken in the morning passes (Dinessen and Hackett, 2011; WMO, 2017). Other satellite sources (optical and passive microwave) may be used when no SAR scenes are available. The ice charts distinguish between six ice classes that are defined by the following thresholds in SIC: fast ice ($SIC == 1.0$), very close drift ice ($0.9 < SIC < 1.0$), close drift ice ($0.7 < SIC < 0.9$), open drift ice ($0.4 < SIC < 0.7$), very open drift ice ($0.1 < SIC < 0.4$), and open water ($0.0 < SIC < 0.1$). They provide a higher-resolution assessment of ice conditions that is independent from the passive microwave data and exhibit different systematic errors.

## 4.5 Passive microwave ice drift vectors

The sea ice drift is monitored using an analysis of subsequent passive microwave images. Data from SSMIS, ASCAT, and AMSR2 sensors are processed using the continuous maximum cross correlation method (Lavergne et al., 2010) and provided in near-real time by the Ocean and Sea Ice Satellite Application Facility (OSISAF). The OSISAF ice drift product represents two-day averages of ice drift on 62.5 km resolution. Model data from *Barents-2.5* is interpolated to the same grid as the observation data and time-averaged over the same time window for comparison.

## 4.6 High-frequency radar surface currents

We use data from a CODAR SeaSonde HF radar operated by MET at the coast of Northern Norway on the Fruholmen island (71.09°N, 23.98°E) to validate the statistical skill of surface currents from *Barents-2.5*. The Fruholmen HF radar operates at a central frequency of 4.453 MHz, and can reach distances up to 180 km with a range resolution of approximately 5 km. We compared hourly-observed HF-radar radial current components with modeled radial current components from the *Barents-2.5* for the period November-December 2021.

## 5 Operational implementation

*Barents-2.5* is executed four times daily, spreading the 24 ensemble members into four sets of six members – as visualized in Fig. 3. While a new analysis by the EnKF is only computed at 00 UTC, each bulletin time benefits from the updated NWP forecasts. In practise, the 00 UTC model run is executed with a six hour real-time delay to allow for enough time for processing incoming observations. The atmospheric forcing from MET-AA is available with four hours delay. The 06, 12, and 18 UTC

model runs are executed with a four hour delay as these only need to wait for the updated atmospheric forecast. The first member in each set is forced by the most recent MET-AA forecast, and the remaining members by ECMWF-ENS.

## 5.1 Ensemble prediction system

At each of the four daily bulletin times, six ensemble members are executed with a 66 hours forecast range. Each member is initialized by the state of the same member from the previous day, with individual analysis increments from the EnKF DA scheme, resulting in 24 unique members. The forecast runs after 06, 12, and 18 UTC require a spin-up run from the analysis time at 00 UTC.

A conscious choice for the design of the ensemble forecast system is to initialize the ensemble runs from the model states of previous-day forecasts, instead of using perturbed states of the identical analysis which is more common in NWP. The latter approach yields an estimate of the uncertainty when the initial condition is well constrained by the observing network and most of the uncertainty arises internally within the forecast cycle.

The *Barents-2.5* EPS, however, retains the state of previous forecast runs in order to preserve sufficient ensemble spread. The EnKF moderates the ensemble spread. Such an approach is beneficial because the initial state of the ocean circulation is poorly observed. At the same time, the mesoscale circulation exhibit features with time scales larger than the assimilation window and forecast range. Therefore, a small perturbation of the initial state would not yield ensemble spread in the forecast that covers the actual uncertainty in the mesoscale circulation.

To represent the actual forecast uncertainties in the circulation field, we initialize the EPS forecast with largely varying initial conditions in the mesoscale circulation. At the same time, the EnKF reduces the ensemble spread for variables that are observed well, i.e., SST and SIC. The ensemble spread for these variables is evaluated in section 7.4.

## 5.2 Operationalisation

*Barents-2.5* is operational in its current setup since September 2021, tagged as version 2.0. The model is implemented at MET and part of Norway's national ocean and weather forecast service. Timely triggering of the model components, as well as pre-processing steps for observation data, forcing and boundary conditions are managed through ECMWF's scheduling software package *ecflow* [10]. Backup measures are in place to secure a reliable operation of the *Barents-2.5* setup to deal with exceptions from normal operation:

- Computing and data storage facilities are set up in two physical locations to mitigate the risk of hardware failures. All model pre- and post-processing is set up in both locations with a continuous synchronization.

- For pre-processing of forcing and boundary data, alternative data sources have been set up, e.g. older forecast cycles from the ECMWF-ENS atmospheric forcing and the *Topaz4* model can be accessed to produce necessary input data.

- Initial states can be obtained from the previous two days forecast cycles of *Barents-2.5*, in case of lost forecast cycles during temporary outages.

---

[10]https://github.com/ecmwf/ecflow, accessed 2023-01-29

– The forecast runs can be executed without analysis increments by the EnKF.

Such exceptions are dealt with automatically through triggering in *ecflow*. If new types of technical errors occur, they are
added to the scheduling system to deal with the same problems in the future. Typical failures have been hardware malfunc-
tioning, broken input data chains, or delays in the observation data chain due to high loads of satellite data during cloud free
conditions. While the first three types of exceptions rarely occur, a failure of the EnKF update is more common. This may occur
either due to unexpected deviations during observation data processing, or due to poorly performing statistic representations
in the EnKF algorithm, e.g. as a result of insufficient ensemble spread or model bias. In such cases, which occur about twice a
month, the model continues without DA increments.

## 6   Validation methods

### 6.1   Validation of sea surface temperature

The upper model layer temperature is validated against SST measurements from the SLSTR sensor of the Sentinel-3A/B
satellites using its Near Real Time (NRT) data product. Only SST data with the best quality flag are used. Multiple swaths in
the model domain may be available each day, but only cloud free conditions provide SLSTR SST measurements. A composite
map of the difference between model values and SST observations within a 24 hours period is shown in Fig. 4. As the coverage
of SST observations varies, the validation of SST is not uniform throughout the day, season or location, providing generally
more data during spring and summer, and during daytime.

In the validation setup, SST swaths are combined into six-hourly fields centered around 00, 06, 12, and 18 UTC. Since
we compare gridded model data with observations at point locations, we upscale the observations to the model resolution by
averaging all observation within a grid cell over the six-hour interval. The model domain is divided into areas as shown in
Fig. 5, which are the same regions as used in validation of the Arctic CMEMS product. These regions exhibit different water
depths, water masses, and ice conditions. We compute the root mean square error (RMSE) and mean error (ME) from the
available observations on a weekly basis to gather enough data for representable statistics. The metrics are computed for the
separate regions and, in addition, an area-weighted mean of all areas is calculated.

The area-weighted mean is also calculated as a function of model lead time, ranging from the 0 hour forecast up to the
48 hour forecast, with an interval of six hours. For comparison, we calculate an RMSE and ME of the persistence forecast,
which is the difference between a past observation with a later observation for the same forecast intervals. By definition, the
persistence forecast yields zero error for the 0 hour forecast. Furthermore, we compute the sum of the observation at the forecast
reference time and the change of SST between the forecast reference time and the forecast lead time, which we refer to as the
model trend. The model trend provides a forecast without an initial error from the model, focusing on the model's ability to
predict changes in SST. Persistence and model trend may only be calculated when observation locations of the model analysis
time overlap with future observation locations for the forecast lead time.

A rank histogram (Hamill, 2001) is calculated from the modeled ensemble SSTs. The objective is to assess whether the spread provided by the EPS matches the actual uncertainty of the forecast system compared to the SST observations. For a given SST observation, the rank of the observation in the model ensemble is identified by finding the number of members with a lower SST value. The rank is hence ranging from $1 < N + 1$, where $N = 24$ is the number of ensemble members. The rank $N + 1$ is assigned if the observation value is larger than all ensemble values. In an ideal ensemble that reflects the uncertainty of the forecast system, each rank number has equal probability of occurrence. Underdispersive ensembles are identified by accumulation of observations in the lowest and highest ranks, meaning that the model values rarely yield as high or low values as seen in observations. A biased model would yield a skewed histogram.

## 6.2 Validation of sea ice concentration

Model SIC is validated against SIRANO SIC, which is used for assimilation, and, in addition, against the subjective ice charts, which are not part of the DA system, in order to provide an independent assessment of the model performance. We validate SIC within an area around the ice edge. After detecting the ice edge (the transition zone between open water and very open drift ice) based on the observed SIC field, we include a zone by a footprint of 150 km around the edge. This method is used to avoid that the total ice extent does not drive the RMSE and ME metrics. At the same time, areas of open water along the sea ice edge are included in the validation. Since model results are continuous values, while ice charts are based on different sea ice classifications (as mentioned in section 4.4), we map the model SIC into the same classes as the ice charts. In order to do an accurate comparison between all available data in the validation, the same classification is used on the SIRANO product. Daily metrics of ME and RMSE are computed for the model analysis, 24 hours forecast lead time, and 48 hours lead time.

An assessment of the ensemble's ability to describe uncertainty in SIC is obtained by means of the reliability to describe probabilities for exceeding certain SIC thresholds (e.g. Saetra et al., 2004; Bröcker and Smith, 2007). The reliability diagram fulfills a similar purpose as the rank histogram for SST, but is more suitable for non-Gaussian distributed variables. The method consists of the following steps: i) A binary event is defined by checking whether SIC is above or below a certain treshhold value, ii) the forecast probability of this event occurring in the model ensemble is assessed for each grid point and time step, iii) forecast probability intervals are mapped onto how often this event occurs in the observations for the same cases.

In a highly reliable prediction system, the observed frequency matches forecast probability for each probability interval, resulting in a straight diagonal line for the reliability diagram that plots observed frequency against model probability. Deviations from the diagonal are due to i) imperfect ensemble spread and ii) model biases. An ensemble with insufficient spread will have higher model probabilities for low observed frequencies, and lower model probabilities for high observed frequencies, resulting in a reversed S-shaped reliability diagram. A normal S-shape indicates excessive model spread. A biased model exhibits too low or too high probabilities for all observation frequencies, resulting in an upwards or downward shifted curve in the reliability diagram (e.g. Bröcker and Smith, 2007).

The method allows for direct comparison with the reported ice classes that are commonly used in ice charting, and we use the same SIC intervals as defined in Sec. 4.4 for calculating the reliability of the ensemble. For validation purposes, open-water and ice-free classes are merged into one SIC class ($0 < SIC < 0.1$) because they are indistinguishable in the SIRANO SIC product.

## 7 Performance of *Barents-2.5*

The general performance of *Barents-2.5* is evaluated: i) by comparison with independent observations not used for DA, ii) in terms of validation metrics close to analysis time (0-24 hours), iii) in terms of skill in the forecast range up to 66 hours, and iv) in terms of its ensemble spread. The spread controls the EnKF DA scheme, and is in addition used to estimate the forecast uncertainty.

### 7.1 Validation against independent observations

Observations that were not assimilated during the model analysis cycles include a subset of the available *in-situ* temperature and salinity data, subjective ice charts, ice drift vectors, and radial HF-radar surface currents.

The performance of *Barents-2.5* to represent the hydrography in the model domain is evaluated in Fig. 6 by means of temperature and salinity, i.e. TS-diagram. The diagram shows the organisation of water masses. In general, the model captures the variability of water masses for the presented data. Errors in temperature are larger for the colder water masses, particularly at depths around 500-1000 m. We also see that the warmer waters above 10 °C are slightly too fresh.

The quality of ocean surface currents in the model is assessed by comparison with radial currents observed by a HF radar antenna at the coast of Northern Norway (Fig. 7). The histogram of radial current speed (Fig. 7a) shows that the model values for radial current exhibit a similar distribution to observations. However, the most extreme velocities are underestimated while low velocities occur too often in the model. The 2D histogram of model speed versus observed speed (Fig. 7b) shows that the model has limited capabilities in terms of predictive skill for surface currents at the observed scale, but they are correlated. In the same figure, the quantile-quantile graph is indicating that the model achieves good statistical skill for the radial surface current speeds.

The sea ice component is validated by the comparison of model SIC with subjective ice charts (Fig. 8) and model sea ice velocities with OSISAF sea ice drift (Fig. 9). The performance in terms of SIC largely depends on the availability of the DA system. Periods without DA result in larger errors for SIC, which is further discussed below. Sea ice drift generally matches the direction of the observed drift (Fig. 9b), which confirms the description of sea ice dynamics in response to ocean currents and winds. However, we note that the ice moves generally too fast in the model compared with the OSISAF ice drift vectors (Fig. 9a).

### 7.2 Validation of model analysis

Validation of model SST analysis using SST from Sentinel-3 SLSTR swaths, is displayed for the course of one year in Fig. 10, for each region separately. Only SST observations from the Sentinel-3 satellite have been used in order to avoid bias differences of various sensor platforms in the validation. Model errors in SST peak during summer months, which is the time period with the most rapid changes in SST and in skin temperature driven by short-wave radiation.

A comparison between observed SIC (ref. Sec. 4.3) with model values is shown in Fig. 11 for an exemplary day. A comparison for the full model domain is possible during most days, given the coverage by the AMSR2 swaths. Daily validation metrics

(RMSE and ME) for SIC in forecast ranges of 0-18 hours, 24-42 hours, and 48-66 hours are computed during the course of a year and shown in Fig. 8.

As a second reference for validation of SIC, a comparison with subjective ice charts is provided in Fig. 8a. The largest anomalies in the model errors for SIC are related to the model spin-up and failures in the DA scheme. During the beginning of the year in consideration, deviations occurred more frequent and during the second half of the displayed period, very

few technical failures of the assimilation scheme occurred, maintaining low model error. Likewise, the model performance deteriorates within a few days when no DA is applied for subsequent cycles, as seen for two periods during March and May 2022 in Fig. 8. A positive SIC bias is present in the model during periods without DA, possibly as a result of too fast ice drift velocities resulting in the ice cover to extend into open water areas.

### 7.3 Forecast skill

Predictive skill in the model forecast is assessed by computing model validation metrics as a function of forecast lead time, e.g. the validation metrics can be obtained by comparison with a value of older forecast cycles. For SIC, we see that the model error is larger for higher lead times, indicating skill in the model analysis (see Fig. 8). In the periods with no DA, the model error grows rapidly and is distinguished in various lead times. Hence we conclude that skill in sea ice forecast is provided by assimilation of SIC.

For SST, we provide a mapping of model error as function of model lead time as averages over periods of three months (Fig. 12). Time averaging of the model errors for many forecast cycles is necessary in this case because individual days rely on the availability of sparse SST observations, covering different regions from day to day as shown for one particular day in Fig. 4.

Comparing SST validation metrics with the persistence forecast as a reference (e.g. Mittermaier, 2008), we describe skill of

445 a model system in two ways:

- The model forecast is considered skillful for lead times where the error of the model forecast is lower than the reference forecast.

- The model analysis is skillful compared to the forecast if the model error grows with lead time.

Skill in model SST is present for forecast ranges beyond 12 hours in spring and summer time, extending beyond the forecast

range of 66 hours (Figs. 12b,c). For the autumn and winter time, we see that the model error is on par with the errors in the persistence forecasts beyond 12 hours lead time (Figs. 12a,d). In general, combining the model trend with past observations performs better at intermediate lead times, but the valuable information content in observations vanishes for longer lead times. In this range, dynamical changes that are resolved by the model provide superior value in predicting the ocean state conditions.

Skill in SIC analysis is concluded from the fact that the short forecast range (0-24 hours) yields lower errors than the

455 longer range forecasts. However, the modeled SIC field exhibit large systematic errors in terms of ME and RMSE (Fig. 8). In particular, SIC increases rapidly in periods without DA. This uncertainty is also present in the difference between the two observation types used for validation, e.g. passive microwave SIC and ice charts. Some forecast cycles show skill compared

to the persistence forecast (not shown), but most cycles do not provide a better forecast than using the last observation as reference. Model skill is present in situations with rapid movements in the sea ice edge that are driven by strong winds. Re-freezing of the sea ice cover in the winter is not as skillful in the model predictions as summer melting and dynamic response to wind forcing.

## 7.4 Ensemble spread

The ensemble of model states is an integral part of the EnKF DA scheme: it provides model error covariances to weigh observational impact with model state in order to compute model increments in each analysis cycle through Eq. 3. The objective of the ensemble spread is to reflect the actual uncertainty of the model, which is possible to assess by comparing observations with the ensemble by means of a rank histogram and reliability (see Sec. 6).

The rank histogram for SST is shown in Fig. 13. We see a fair ensemble spread for this variable across the center part (i.e. a flat distribution in the rank histogram), but note a skewed offset that reflects a negative overall bias in model SST. We also acknowledge that the most extreme SSTs are not represented in the model, indicated by excessive lowest and highest ranks.

In Fig. 14, the spread in SIC is evaluated using a reliability diagram. We assess the reliability of forecast probabilities for occurrences of open water up to a given ice class (Fig. 14a). The opposite approach assesses forecast probabilities of high SICs down to a certain ice class minimum (Fig. 14b). Both figures indicate an offset that represents a positive SIC bias of the model, e.g., forecast probabilities for SIC tend to be too low from the ensemble. In addition, the reversed S-shape in both figures indicates too low ensemble spread. Parts of the insufficient ensemble spread seen in the reliability diagram could be attributed to observation errors, as discussed in (Saetra et al., 2004), but not all. The highest forecast probabilities of open water, e.g., 80% chance for the occurrence of $0 < SIC < 0.4$ is nevertheless addressed fairly well by the model. Also, low chances of occurrences for high SIC, e.g., in the range $0.4 < SIC < 1$, are predicted accurately by the model. Addressing the lack of sufficient EPS spread, using forecast probabilities in decision making could benefit from scaling as a post-processing step to match the observed occurrence frequencies (e.g. Chang et al., 2015).

## 7.5 Analysis increments

The EnKF provides new initial states for each ensemble member. For each state variable, the analysis increment is the difference between the new model analysis provided by the EnKF and the model background, i.e. the forecast of the previous cycle. The objective of the analysis increments provided by the EnKF is to reduce deviations from the observation while maintaining ensemble spread to reflect uncertainty. In this section, we show some characteristics of the analysis increments for an exemplary day in order to illustrate the functioning of the DA scheme. Similar model increments are calculated for the analysis in each forecast cycle, applied daily at 00 UTC.

Fig. 15 shows forecast increments of SST, surface salinity, and SIC on 2022-12-15. The differences result from the observations that were available during the preceding 24 hours. The most substantial increments in model hydrography occur in regions where both SST and *in-situ* observations are present (see Fig. 2) and spread in SST is large (Fig. 15d). *In-situ* observa-

490 tions provide rather localized increments in surface fields. SIC increments are most pronounced in the marginal ice zone where the ensemble has large spread (Fig. 15d).

To confirm that the DA increments move the model state towards the observed ocean state, we present a binned scatter histogram of observed vs. model values for SST and SIC in Fig. 16. The correlation between modeled and observed SST is slightly improved for the analysis (right panel) compared to the background (left panel). In the case of SIC, which generally

has larger errors than SST, we see more radical updates to the model state in each cycle.

Ensemble spread - which is the standard deviation computed from the ensemble of each state variable – is an essential characteristic of the EPS, and sufficient ensemble spread is required by the EnKF to yield appropriate model increments. Spread increments, i.e. the difference between ensemble spread before and after DA, is generally negative and the spread again grows during the model forecast.

The spread for SST, salinity, and SIC during one analysis cycle is shown in Figs. 15d-f. Spread in SST and surface salinity is present throughout the model domain, however pronounced in certain areas, e.g. below the sea ice cover. This indicates pronounced uncertainty in the model state, e.g. the present hydrography below the sea ice cover is not well known and the model is expected to exhibit larger errors in these areas.

The ensemble spread is reduced in each analysis such that model state variables converge towards the observations. The

505 spread increment is shown in Figs. 15g-i. For SST, we see a substantial reduction in spread that is associated with coverage by SST observations during that analysis cycle. All shown variables experience a spread reduction in the marginal ice zone, owing to the impact of SIC observations in this area. There is little modification to ensemble spread below the dense sea ice cover and in other places where no observations are present during the analysis cycle.

Model state variables that are not observed are adjusted by the EnKF along the observed variables, based on the covariance

matrices between each variable. This includes the 3D current field, surface elevation, and all ice state variables. Surface current increments are at the order of 0.05 m/s and sea surface elevation increments are at the order of 0.01 m and mostly associated with alterations in the density field due to modifications in temperature fields.

## 8   Discussion

The *Barents-2.5* ocean and sea ice forecast model provides daily short-term predictions with a forecast range up to 66 hours.

This operational model suite has been built from various components and code repositories; the dynamic ocean component is based on ROMS v3.7 (Shchepetkin and McWilliams, 2005), the dynamic sea ice component is CICE 5.1 (Hunke et al., 2017), model coupling is achieved by MCT (Larson et al., 2005), DA is handled through *EnKF-C*, and operational scheduling is implemented in *ecflow* including the maintenance of the ensemble. In addition, *Barents-2.5* involves static data such as the GEBCO bathymetry and TPXO tidal constituents, and a constantly updated stream of forcing data. The atmospheric forcing

is provided by ECWMF-ENS and MET-AA, lateral boundary conditions are provided by *Topaz4*, and river influx is generated from climatology and data from the Norwegian Water Resources and Energy Directorate. Finally, observations of SIC, SST,

and *in-situ* hydrography are assimilated. All components combined constitute the *Barents-2.5* model, and the model capabilities as well as the presented validation are a consequence of the interactions between all those components.

## 8.1 Forecast capabilities

Forecast skill and validation metrics have been presented for SST and SIC. These are central variables describing the state of ocean and sea ice conditions that are frequently observed by satellite imagery. The forecast skill and ensemble spread for surface currents are the subject of a separate investigation in Idžanović et al. (2023).

*Barents-2.5* shows predictive skill in temperature forecasts (Fig. 12) compared to the persistence reference by Sentinel-3 SST imagery. Forecast skill is better during spring and summer time when SST observations are more abundant and solar

radiation drives rapid changes in SST. Mismatches in the modeled ice cover dominate the SST errors in the marginal ice zone, therefore all regions that contain ice have been excluded in the consideration for SST validation. SST fields are directly needed in forecast applications, e.g. to warn about icing on ships (Samuelsen, 2018) and to provide surface forcing in NWP models (e.g. Müller et al., 2017).

Skillful SST predictions in the forecast range are a consequence from the models capability to represent realistic air-sea

fluxes, vertical mixing, and water mass transport. A requirement for low errors at the analysis time is the successful constraint by observations. Without DA, the model state tends to drift away from the real ocean state, owing to compounding of systematic model errors. Such model state drift is particularly evident for SIC, during the periods when no DA is applied (Fig. 8) owing to a systematic model bias in SIC (Figs. 8,11). Due to varying SST observation availability, SST validation is not comparable between daily cycles, but similar drift may occur. For SIC, we see that the model state looses skill within a few days without

DA. Deterioration of model skill within a few days is generally expected when DA is halted in regional ocean models (e.g. Moore et al., 2011).

Validation of SST indicates larger errors during summer time (Fig. 10) but also a higher skill compared to the persistence forecast (Fig. 12). Variability in SST during the summer time is generally higher, resulting from the availability of direct sunlight to heat up the surface. Most of the model region is in high latitudes with midnight sun. The diurnal effect may dominate

in early spring time, while in the summer the solar heating is largely controlled by cloud cover. Skillful SST predictions by the ocean model thus require accurate cloud cover and atmospheric radiative fluxes, highlighting the role of the used NWP model. Due to the stronger solar forcing, SST in summer time changes more rapidly than during the winter season. *Barents-2.5* is forced by hourly fields from MET-AA, giving SST forecasts an advantage over the persistence forecast during spring and summer (Fig. 12).

While errors in SIC are variable throughout the year (Fig. 8), we see a persistent advantage of short lead times compared to longer lead times. The model is hence skillful, but the remaining systematic model errors make it difficult to use predicted SIC fields directly in forecasting applications. Use of ice charts based on remote sensing alongside model forecast remains necessary. Predictions of ice drift in ocean-ice forecast models have been shown to be skillful (Schweiger and Zhang, 2015), resulting from the skillful NWP and realistic description of sea ice rheology. The freezing and melting along the ice edge,

however, depends on a number of thermodynamic processes (Fichefet and Maqueda, 1997), and these are critically sensitive to water mass properties and higher resolution details in the atmospheric forcing (Kusahara et al., 2017).

## 8.2 Configuration of the ensemble and data assimilation schemes

The information provided by the EPS is useful in two ways. Firstly, ensemble spread guides the DA scheme. Secondly, the ensemble allows users to assess model uncertainty in the forecast range. Spread in SST is satisfactory (Fig. 13), with only slight
underestimation of extremes. Spread in SIC is too low (Fig. 14), possibly as a result of a positive SIC bias, but the model shows a capability to predict the uncertainty in ice forecasts.

Difficulties in representing SIC spread in similar EnKF setups have previously been reported by Lisæter et al. (2003). As a bounded variable between 0 and 1, SIC is generally non-Gaussian but the EnKF formerly requires Gaussian distributed variables and observation errors. Fig. 17 shows some examples of local SIC distributions across the ensemble. We identify that
SIC may be approximated by Gaussian statistics for intermediate SIC, but close to areas of dense ice (SIC$\rightarrow$1) and open water (0$\leftarrow$SIC) we notice that SIC exhibits skewed and bounded distributions, which are occasionally bi-modal. We hence expect a weakness in the applied DA method in areas close to open water or very dense ice. Methods to address these issue are provided by e.g. Bishop (2016) and Anderson (2022). Use of bi-modally distributed variables in DA filters have been discussed in Chan et al. (2023). These techniques are most likely to be encompassed in future model versions of *Barents-2.5*.

The initialisation of the ensemble in each forecast cycle is based on an approach where the individual members retain their identity. DA increments are provided individually for each member and the spread is reduced during each analysis. The major differences between the ensemble members result from diverging history of the members. Each member is also forced by different atmosphere ensemble members, but the most important spread in the mesoscale ocean circulation stems from the history of the EPS.

The ensemble spread is modified through i) reduction by EnKF analysis step, ii) increase during the model integration, and iii) inflation of the analysis increments during the DA analysis. The inflation allows us to control the ensemble spread and avoid collapse. A moderate inflation factor is used for most variables, and slightly higher for SIC which suffers low spread. While higher inflation has the potential to further improve the spread in SIC, we also experience that large inflation factors lead to multivariate artifacts in the model analysis. Anderson (2009) propose a spatially and temporally varying adaptive inflation
algorithm to allow for a larger covariance inflation, leading to a more efficient increase of ensemble spread. El Gharamti (2018) iterates on this method, improving the stability of the adaptive inflation to the occurrence of negative and physically intolerable inflations.

The EPS and EnKF system requires spin-up time, due to the system's dependence on its own history of ensemble spread. Initialisation of the EPS took place in March 2021, and the EnKF was activated in September 2021. About three months were
585 required for a sufficient spin-up of the full DA system.

The choice of a 24 hours analysis cycle is based on the practical need to issue daily forecasts. The assimilation cycle length is a compromise between the amount of observations, the ensembles ability to maintain spread, and the need for rapid update cycles. The number of ensemble members is limited by computational resources. The EnKF DA scheme relies on a statistical

description of possible model states compared to the available observations, and therefore the ensemble needs to cover the degrees of freedom for the ocean state within a localisation radius (Tab. 4), which is maintained in the current setup using 24 ensembles. Only horizontal localisation is applied here, but we note that the system could benefit from vertical localisation as well.

Observations are directly mapped to model state variables in *Barents-2.5*, which is straightforward for *in-situ* temperature and salinity. For SIC, a mapping to aggregated SIC is needed because CICE operates with SIC for separate thickness classes. For SST, the upper model layer temperature is selected to represent the surface temperature, which is a valid assumption during moderate or high winds, cloudy skies, or low solar radiation as most common in Arctic waters. During conditions with low wind and strong solar radiation, the skin temperature retrieved by the satellite sensor differs from the temperature of the upper model layer (Price et al., 1986). Hence we expect discrepancies primarily during spring and summer months, affecting both assimilation and validation routines, which motivates ongoing work on asynchronous SST assimilation using observation operators to describe the skin temperature (e.g. Zeng and Beljaars, 2005).

The current DA setup in *Barents-2.5* evaluates all observations at analysis time 00 UTC. For slowly varying model state parameters, such as the hydrography at depth, this assumption bears limited consequences as long as the model error is larger that the diurnal variability of the observed parameter. Ongoing development work on *Barents-2.5* focuses on assimilation of SIC and SST observations at the observation time, showing positive improvements for modeling SIC [11].

## 8.3 Integration with contingency models

The *Barents-2.5* model forecasts are routinely used in trajectory models that serve as decision support tools in emergency response situations. The OpenDrift framework (Dagestad et al., 2018) encompasses oil spill transport, ice berg drift forecasts, and leeway drift (e.g. person-in-water, large floating objects) applications. All of these models require timely computation of trajectory forecasts that are based on surface currents, wave, and wind forecasts. Among these forcing variables, surface current forecasts exhibit by far the largest uncertainty (Dagestad and Röhrs, 2019). OpenDrift makes use of the *Barents-2.5* EPS by issuing ensembles of trajectory forecasts, allowing one to assess the uncertainty of trajectory simulations in a given situation. In practice, the difference in trajectory forecasts from various ensemble members varies from case to case. The *Barents-2.5* model is useful for distinguishing cases of low and high uncertainty in drift applications (de Aguiar et al., 2023).

## 8.4 Model development history and outlook

An earlier implementation of the *Barents-2.5* has been operational since March 2019, with DA scheme for SIC using a combined optimal interpolation and nudging scheme (Fritzner et al., 2018), which allows the model to reflect the observed SIC state closely. No ocean temperature data was assimilated in the v.1 model implementation, and as a consequence the model hydrography drifted away from the observed ocean state. The most notable changes for the model version described here are the implementation as an EPS with 24 members and the introduction of the EnKF DA scheme for both sea ice and ocean variables.

---

[11]Duran Moro, M. et al. Assimilation of SIC satellite swaths versus daily means in a coupled ocean-sea ice regional model. Submitted to The Cryosphere.

Asynchronous assimilation of SIC, i.e. swath data, is in a development stage and SST assimilation will also likely benefit from asynchronous assimilation. In asynchronous DA, each satellite swath is compared to the model field at the respective retrieval time instead of applying observations at a fixed time of the day. This may lead to a reduction of the representation error in observations and hence introduce a tighter constraint on the model state. Constraint of mesoscale ocean current fields beyond assimilation of SST is planned by assimilation of sea level anomaly from satellite altimetry and HF-radar currents.

The lack of ensemble spread in SIC is a major weakness in the current model setup. A more explicit perturbation of sea ice variables is planned in order to introduce larger spread in SIC, as well as and adoptive inflation in the EnKF analysis (Anderson, 2009; El Gharamti, 2018). Perturbation may be applied to the initial conditions, or as a physics perturbation in the sea ice component by variation of empirical parameters in CICE. In addition, applying transform operators for SIC, e.g. through remapping of SIC to a Gaussian variable, can mitigate difficulties of the EnKF to assimilate SIC directly (e.g. Bishop, 2016; Chan et al., 2023; Cipollone et al., 2023).

A stronger coupling with NWP models and wave prediction models is foreseen as soon as such coupling is shown to improve forecasts. Atmosphere models require SST as lower boundary conditions, and these may benefit from skillful predictions of SST by *Barents-2.5*, provided that the coupling will not introduce artifacts through poorly defined feedback loops. At present, MET-AA, which provides surface forcing for *Barents-2.5*, uses static SST fields based on the Operational Sea Surface Temperature and Ice Analysis (OSTIA) daily satellite retrieval product. Coupling can either be implemented as fully coupled online or by using SST forecasts of the previous forecast cycle. A coupling with wave prediction models is foreseen, in which the wave prediction model provides surface wave dissipation, momentum fluxes and Coriolis-Stokes forcing to the ocean model (e.g. Breivik et al., 2015). Such coupling has been shown to improve SST predictions in the global ocean forecast model at the ECMWF (Janssen, 2012).

*Code and data availability.* A source code for the coupled ROMS/CICE ocean and sea ice model is available through https://doi.org/10.5281/zenodo.5067164 tagged as version 0.4.1 of the METROMS repository. Specific configuration and grid files for *Barents-2.5* are archived at https://doi.org/10.5281/zenodo.7607191. The source code for the EnKF-C v.2.9.9 is obtained from https://github.com/sakov/EnKF-C.git, commit *7eea4d8* as of Jul 8, 2021. The source code for the *ecflow* scheduling software is available at https://github.com/ecmwf/ecflow.

Archived data from the operational model runs of *Barents-2.5* is disseminated on https://thredds.met.no/thredds/fou-hi/barents_eps.html. Observations of satellite SST and *in-situ* hydrography are retrieved from CMEMS https://doi.org/10.48670/moi-00036. Observations of SIC are available through https://thredds.met.no/thredds/osisaf/osisaf_seaiceconc.html . The subjective ice charts data are available as graphic maps and trough an API at https://cryo.met.no/en/latest-ice-charts. The low resolution ice drift observations are available through https://osi-saf.eumetsat.int/products/osi-405-c. Data from the Fruholmen HF radar are accessible in near-real time through https://thredds.met.no/thredds/catalog/remotesensinghfradar/catalog.html.

*Author contributions.* Model setup and implementation: JR, JB, ER, SF, NMK, KW, JD, KHC. Observation preparation: AKS, TL, MI. Configuration of DA scheme: SF, JR, MDM. Model validation: YG, ER, MDM, MI, AKS. Manuscript preparation: all authors. Project management: JR, KHC

*Competing interests.* There are no competing interests associated with this work.

*Acknowledgements.* We acknowledge funding by the Research Council of Norway, grant 237906 (CIRFA), 300329 (EcoPulse), 302917 (SIRANO), 314449 (ACTION), and the Norwegian FRAM Flagship program project SUDARCO (No. 2551323). This study has been conducted using E.U. Copernicus Marine Service Information, i.e. model data as boundary condition and observation data used in assimilation. We acknowledge the R&D contribution of ESA CCI Sea Ice and EUMETSAT OSI SAF projects to the preparation of the regional SIC product. We like to express our gratitude to the reviewers who helped to improved this manuscript and also provided valuable ideas for the 660 future development of the model.

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

| P | M | N | Kmin | Pmin | CMU0* | C1 | C2 | C3M* | C3P | SIGK | SIGP |
|---|---|---|---|---|---|---|---|---|---|---|---|
| 2.0d0 | 1.0d0 | -0.67d0 | 1.0d-8 | 1.0d-8 | 0.5270d0 | 1.0d0 | 1.22d0 | 0.05d0 | 1.0d0 | 0.8d0 | 1.07d0 |

**Table 1.** Parameters for the GLS turbulence scheme used in *Barents-2.5*. *) parameters associated with the choice of stability function, which is set using the `Canuto_A` pre-compiler option.

| Variable | Numeric scheme |
|---|---|
| free surface | Chapman exlicit |
| 2D momentum | Shchepetkin |
| 3D momentum | oblique radiation and nudging |
| salinity and temperature | oblique radiation with nudging |
| TKE and GLS | gradient |

**Table 2.** Numeric boundary solution schemes used in the ocean component for *Barents-2.5*. All four model boundaries use the same schemes. Nudging towards the boundary *Topaz4* fields is imposed within an area of 40 grid points from the boundaries with decaying nudging coefficients.

| Constituent | K2 | S2 | M2 | N2 | K1 | P1 | O1 | MN4 | M4 | MS4 |
|---|---|---|---|---|---|---|---|---|---|---|
| Period [h] | 11.96723 | 12 | 12.4206 | 12.65835 | 23.93447 | 24.06589 | 25.81934 | 6.269174 | 6.210301 | 6.103339 |

**Table 3.** Tidal constituents used for tidal forcing in *Barents-2.5*.

|  | SIC | SST | *in-situ* temperature | *in-situ* salinity |
|---|---|---|---|---|
| $R$ | 10 | 10 | 4 | 10 |
| $r_{loc}$ | 50 km | 50 km | 80 km | 80 km |
| Platform | AMSR2 | AVHRR, VIIRS, SLSTR | profiling floats, drifting buoys, moorings, CTD casts, ships | |

**Table 4.** Observation platforms of assimilated variables and configuration parameters for observation types in EnKF-C.

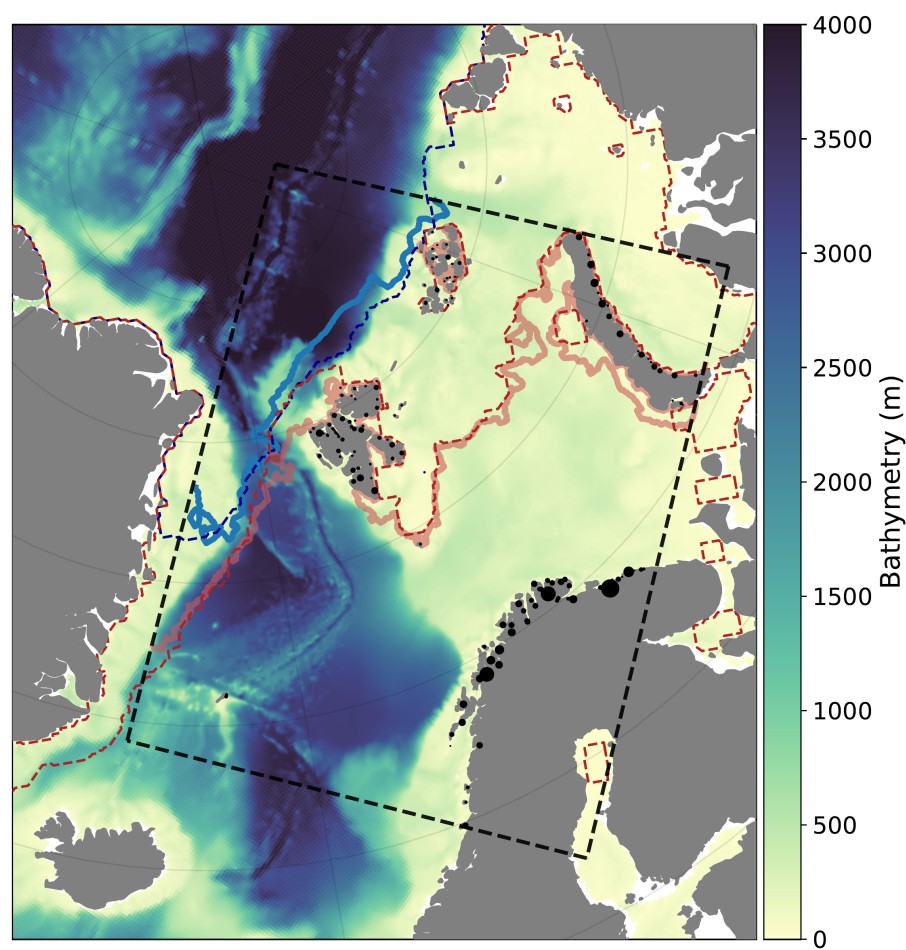

**Figure 1.** Bathymetry of *Barents-2.5* plotted within *Topaz4*, explicitly showing the *Barents-2.5* domain. The locations of river input are marked by black dots, with dot sizes scaled as the yearly mean of river transport. The sea ice edge, here represented by 15% SIC, is shown at the time of minimal (blue) and maximal (red) sea ice extent for each model (*Barents-2.5* in solid lines, and *Topaz4* in dashed lines).

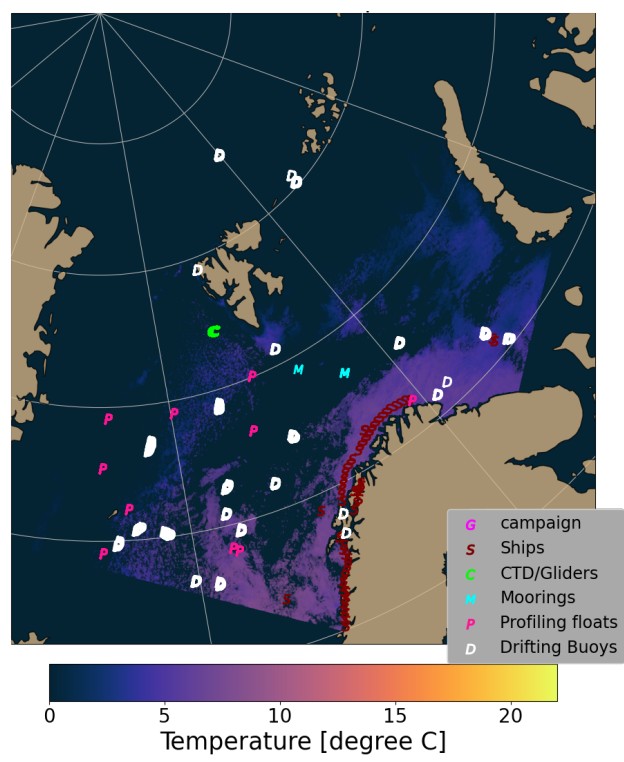

**Figure 2.** *In-situ* data coverage during the assimilation cycle on 2022-12-15. The type of in-situ platforms are indicated in capital letters. The color coded pixels show SLSTR SST observations available during the particular day.

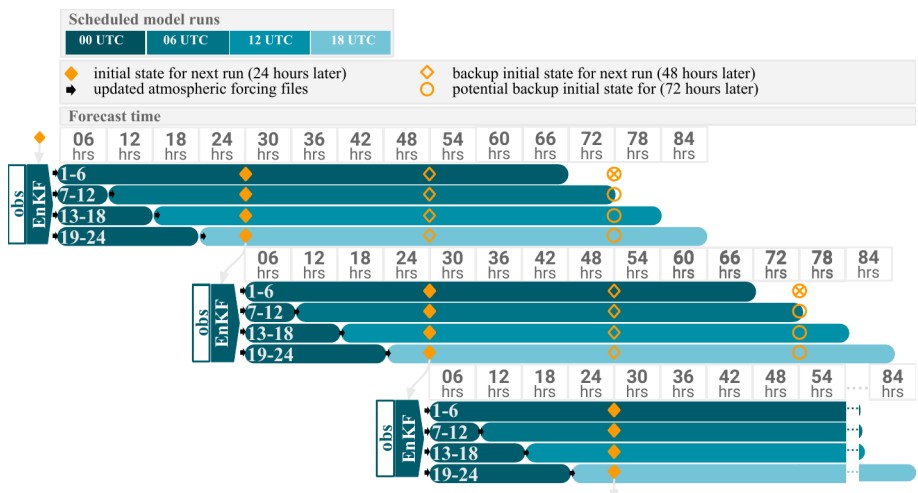

**Figure 3.** Work flow diagram for operational ensemble prediction system. Each horizontal bar represents a group of six individual model runs. At each 00 UTC bulletin time, the EnKF DA scheme provides an analysis as new initial conditions for all 24 members. Every subsequent bulletin time (06, 12, and 18 UTC) includes a short spin-up run from the analysis time at 00 UTC and a forecast run of 66 hours using updated atmospheric forcing.

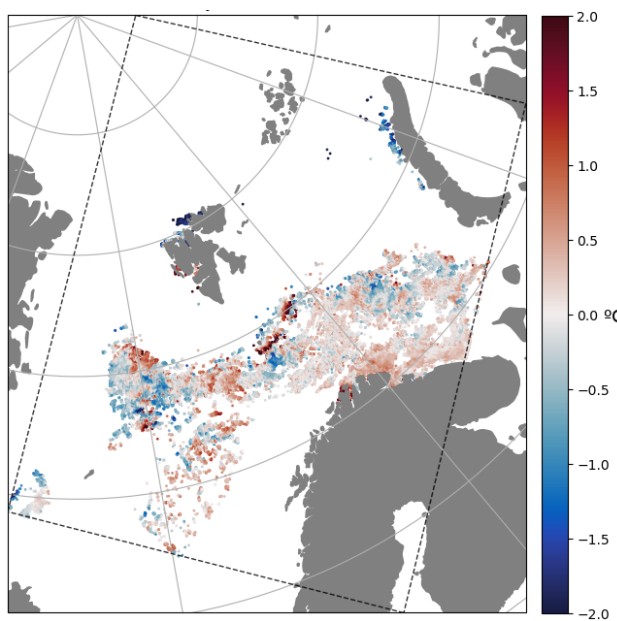

**Figure 4.** Difference between model SST and observed SST from Sentinel-3. The displayed data includes all swaths that pass the model domain during a 24 hours period (2022-12-15). The respective model values for computing differences are taken from the satellite retrieval time.

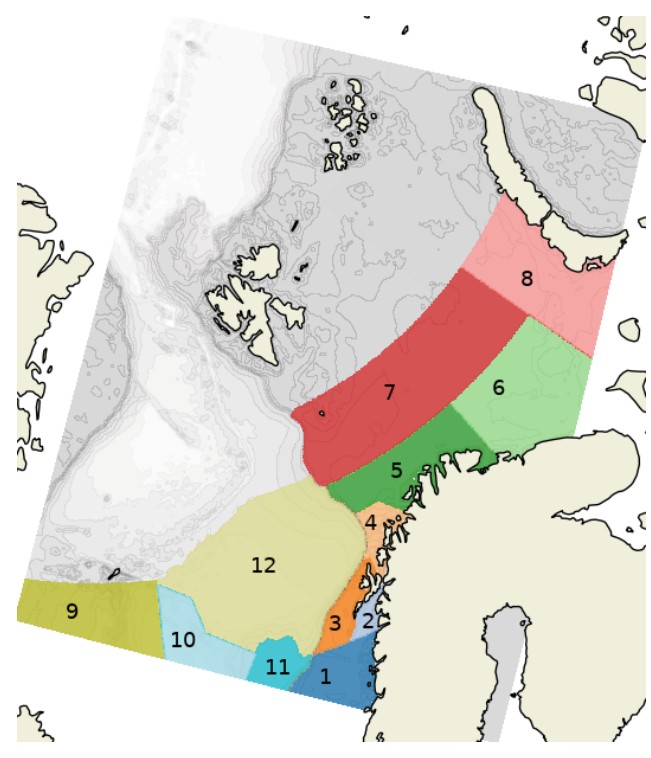

**Figure 5.** Definition of regions used in SST validation. 1) Helgeland Coast, 2) Vestfjorden, 3) Lofoten Vesteralen North, 4) Troms Coast, 5) Barents Sea Southwest, 6) Barents Sea Southeast, 7) Barents Sea Central, 8) Barents Sea East, 9) Nordic Seas Southwest, 10) Nordic Seas Southeast, 11) Voring Plateau, and 12) Lofoten Basin. Areas that include the marginal ice zone (gray) are excluded from the SST validation.

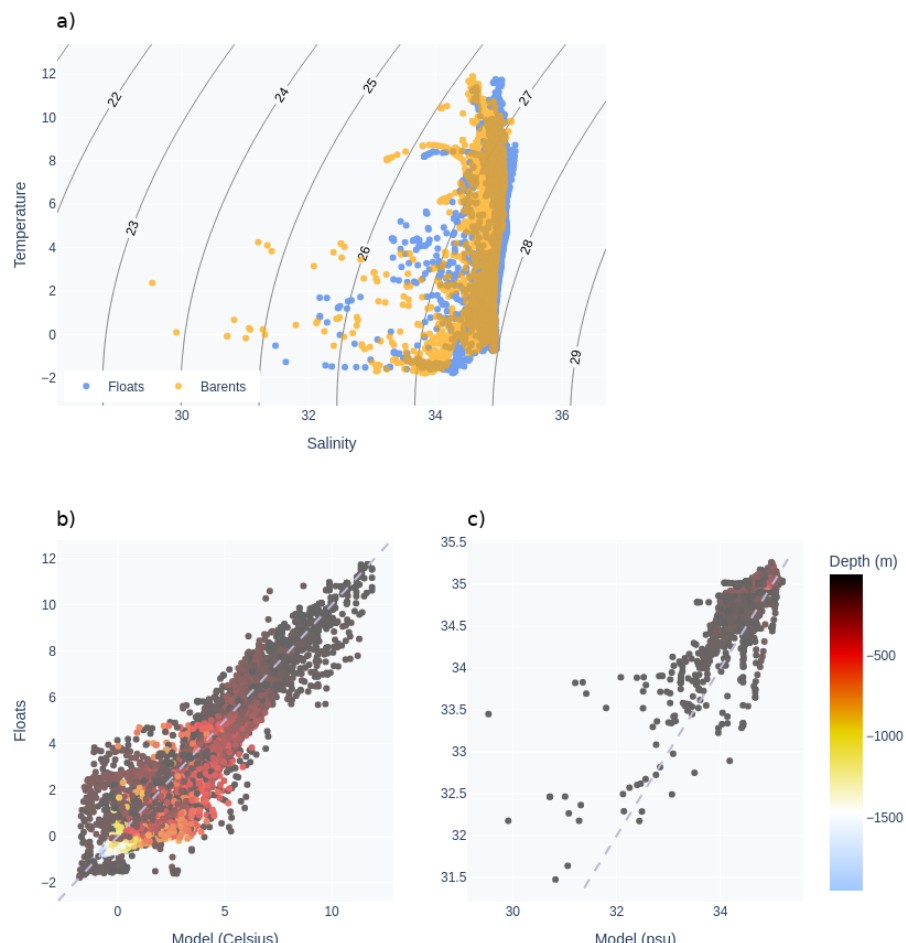

**Figure 6.** Validation of model hydrography against non-assimilated *in-situ* observations from floats in the model domain. a) TS-diagram with density isopycnicals as solid lines. Line labels denote density deviations from $1000\,\text{kg/m}^3$. b) Scatter plot of model versus observed temperature. b) Scatter plot of model versus observed salinity. c) The color coding in (b) and (c) denotes the depth below the sea surface of the data points. The data originate from model periods within in June-November 2022, for dates when the DA scheme did not assimilate data due to technical failures in the DA system.

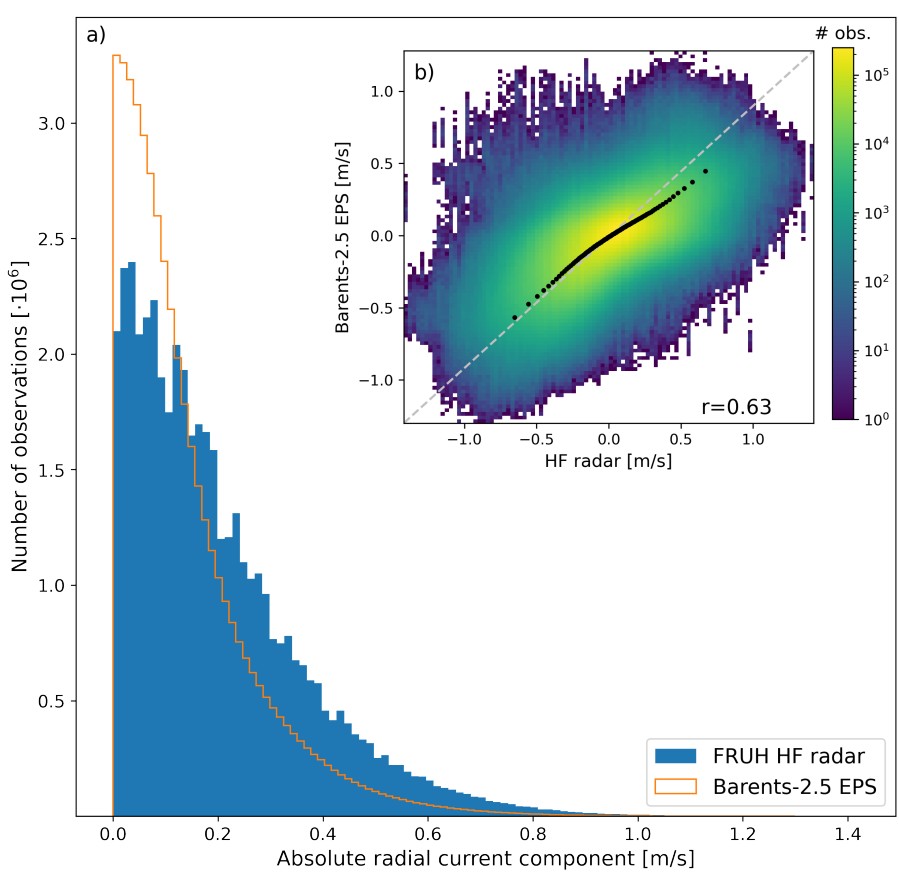

**Figure 7.** a) Distribution of absolute radial current components at Fruholmen from HF-radar observations and *Barents-2.5* EPS. b) 2D histogram and quantile-quantile plot of modeled versus observed radial current components. The vertical axis shows modeled radial current components, while the horizontal axis shows observed HF-radar radial current components. The data presented in both figures correspond to a forecast lead time of 0-24 hours for the period from 2021-11-15 to 2021-12-31.

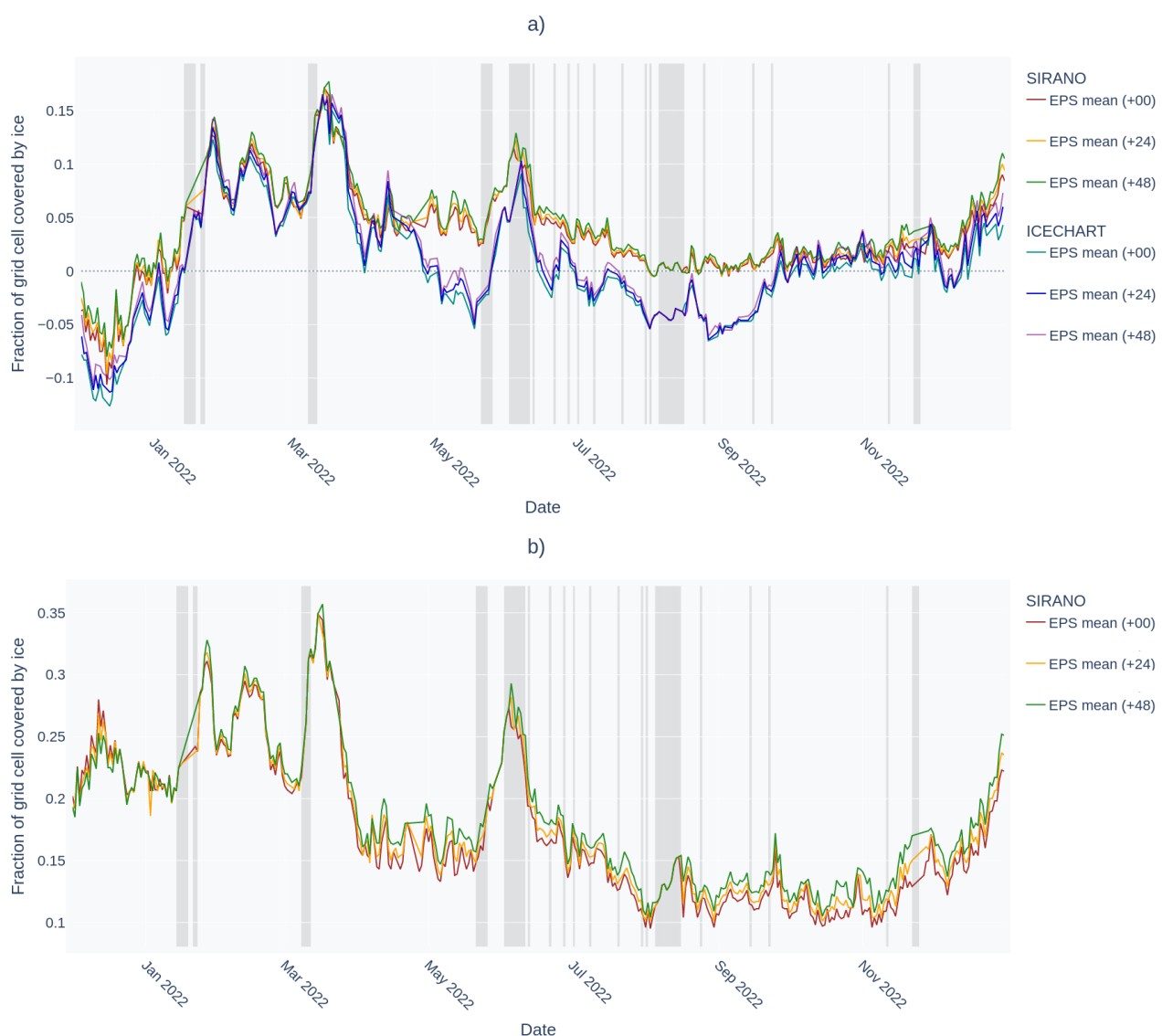

**Figure 8.** Time evolution of (a) ME and (b) RMSE for SIC based on daily comparison of model values in various forecast ranges averaged over the entire model domain. Differences are shown for comparison with the passive microwave SIC product that is assimilated (SIRANO) and for a comparison with subjective ice charts that are produced by ice analysts. Shaded intervals mark model cycles when no data was assimilated.

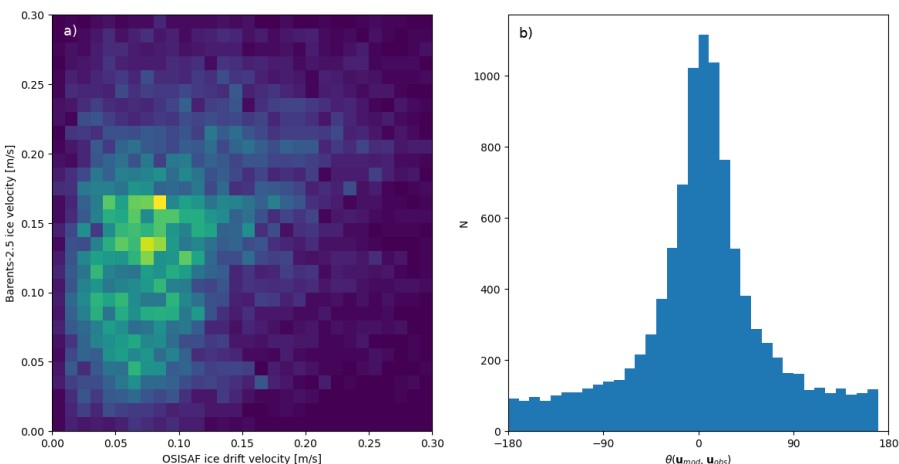

**Figure 9.** Comparison of ice drift velocities from *Barents-2.5* against OSISAF ice drift vectors derived from passive microwave imagery. a) 2D histogram of total ice drift speed. b) Histogram of direction difference $\theta$ between modeled ice drift $u_{mod}$ and observed ice drift $u_{obs}$. The ice drift velocities represent 48 hours averaged velocities.

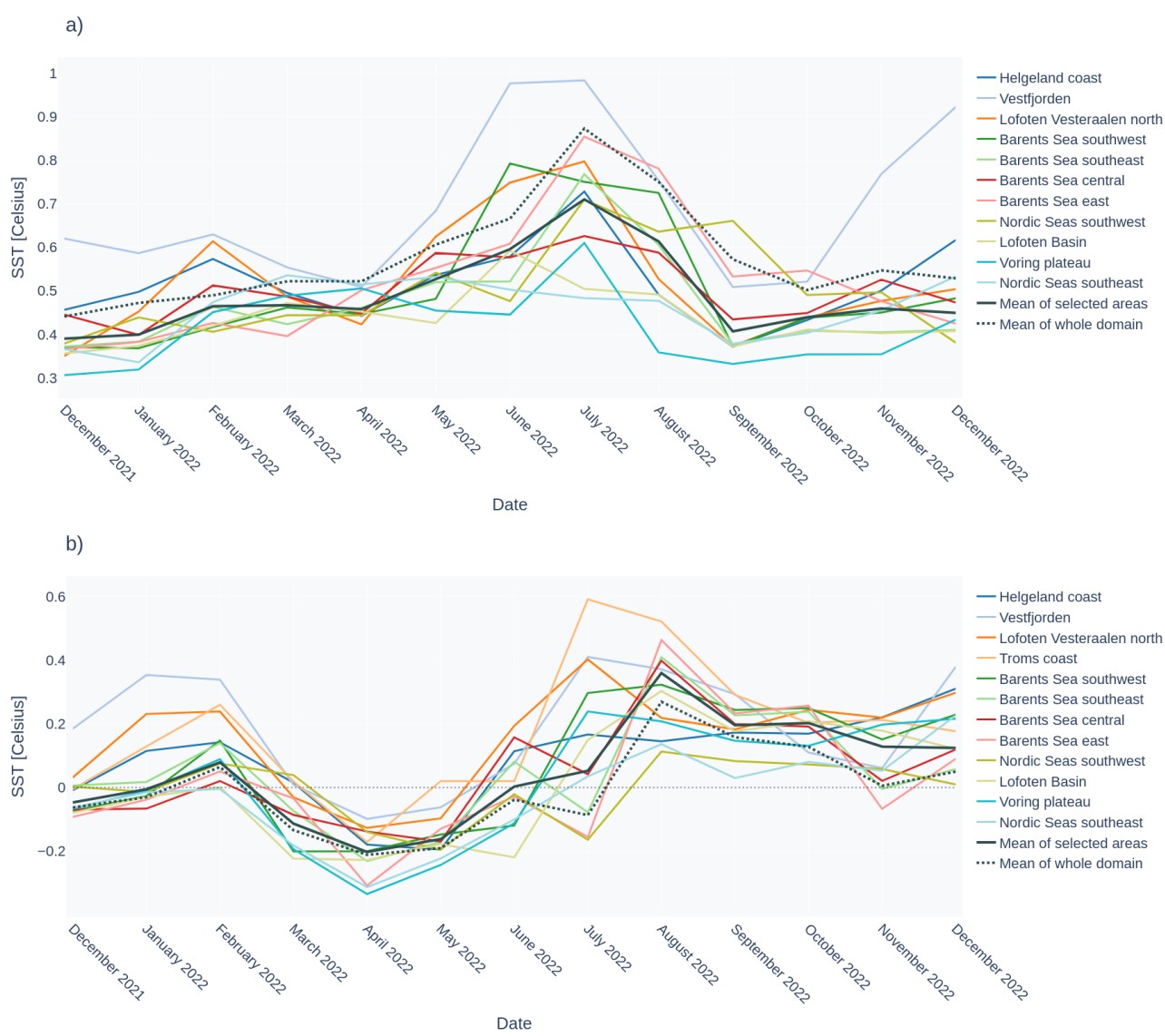

**Figure 10.** Time evolution of (a) RMSE and (b) ME for SST, separated for various regions in the model domain as defined by the colours in Fig. 5. For each available swath, Sentinel-3 SLSTR observations are compared with the model value that is closest to the satellite retrieval time in the 0-24 hours forecast range. Monthly averages of available data are shown.

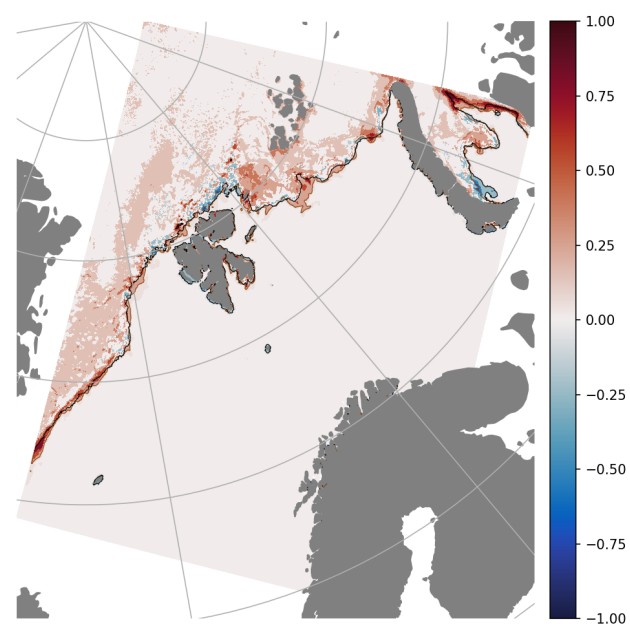

**Figure 11.** Difference between modeled and observed SIRANO SIC on 2022-12-15.

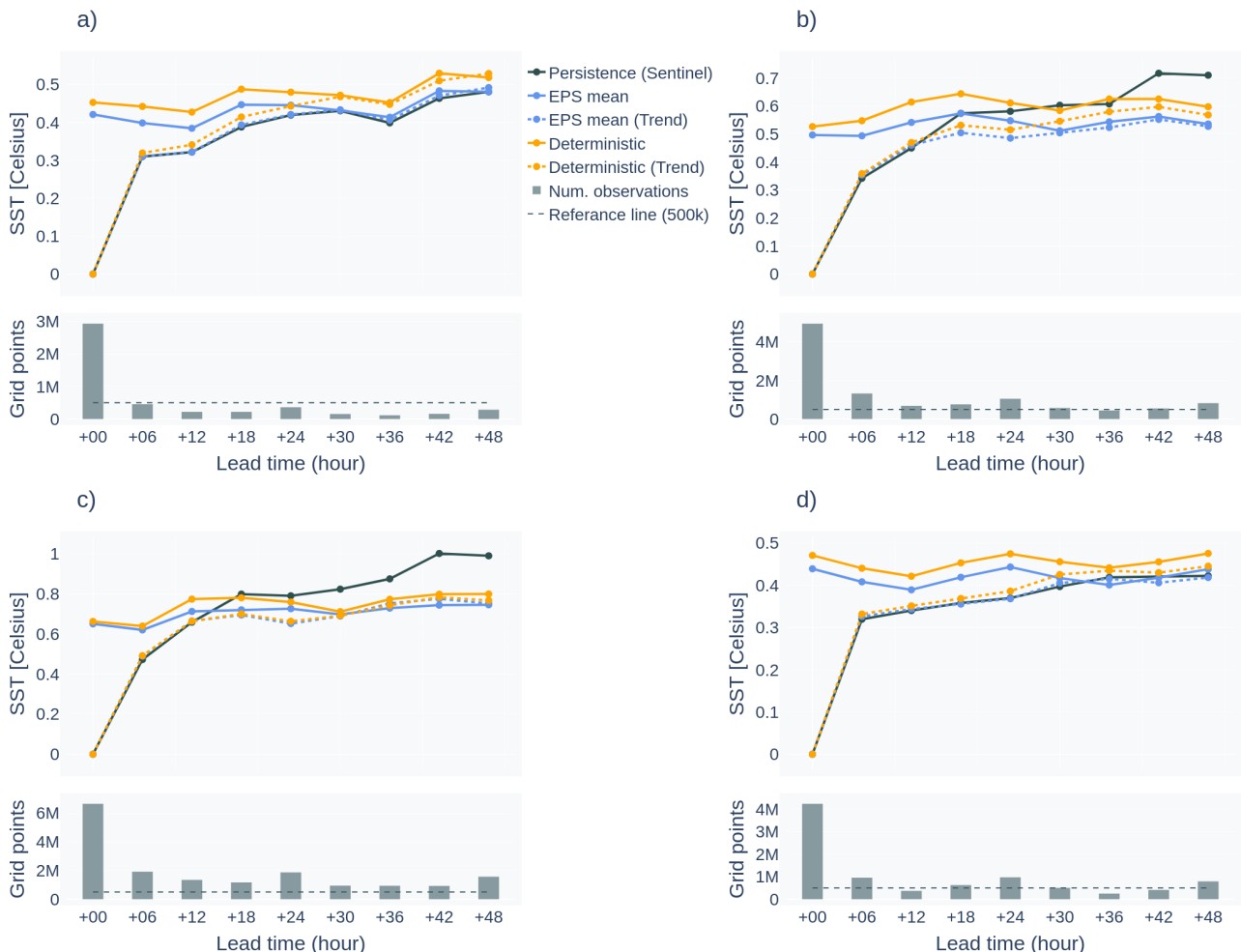

**Figure 12.** RMSE of SST as function of lead time for each season, for a) December-January-February (DJF), b) MAM, c) JJA, d) SON. The black line shows the skill of the persistence forecast, which is root mean square difference between overlapping scenes of SST observations that are shifted by the respective lead time. The dashed lines indicate the skill of the model trend. The bar plot below each panel shows the number of overlapping observations used for the computation of the persistence and model trend skill.

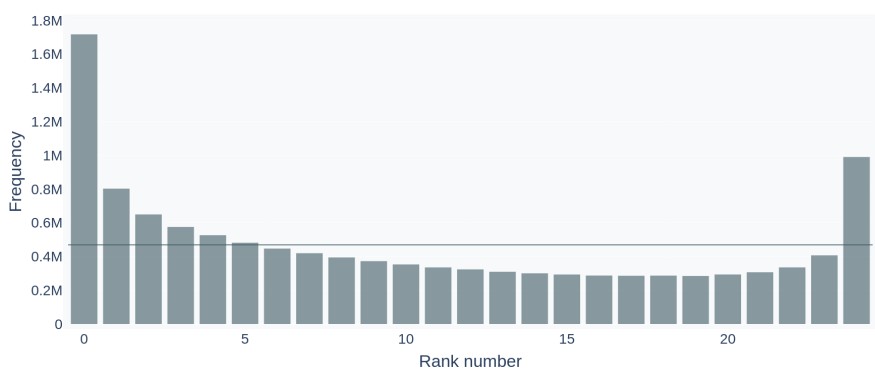

**Figure 13.** Rank histogram for model ensemble SST data during the period from 2022-05-01 to 2022-12-20. Observation rank number refers to the number of ensemble members that have a lower SST value than the corresponding SST observation. Occurrence frequencies are shown in quantities of millions. The horizontal line indicates expected frequencies for an unbiased ensemble.

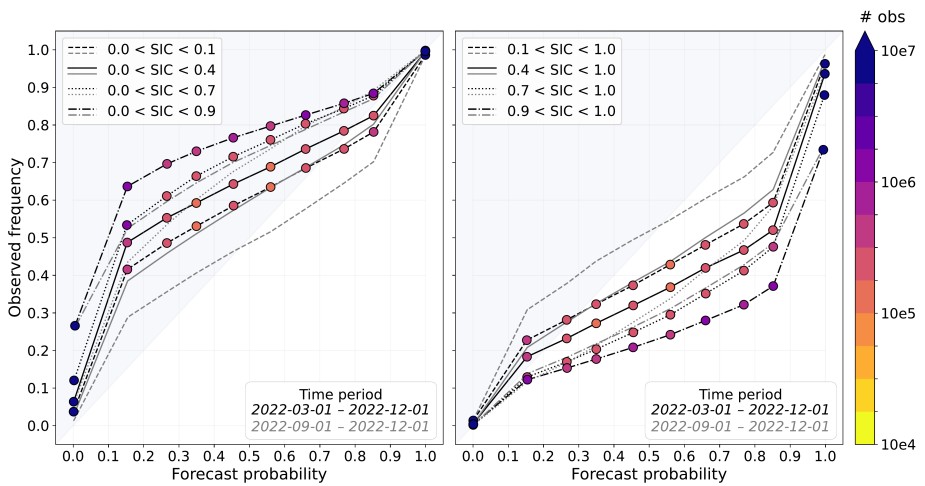

**Figure 14.** Reliability diagram for SIC forecast probabilities derived from the EPS, computed for the period of 2022-03-01 to 2022-12-01 (black lines) and 2022-09-01 to 2022-12-01 (gray lines). The two panels show how well the ensemble system predicts the chance of low SIC occurrence (left panel) and high SIC occurrence (right panel). Forecast probabilities have been computed for specific concentration intervals that are given in each panel, with a lower and upper threshold for SIC. The forecast probabilities calculated for each interval are grouped into evenly spaced bins from 0.0 to 1.0 on the x-axis. The figure shows the mean of the forecast probability within each bin relative to the observed frequency in the same data points. The mean values are highlighted with colors for the period from 2022-03-01 to 2022-12-01, representing the number of observations used in calculations.

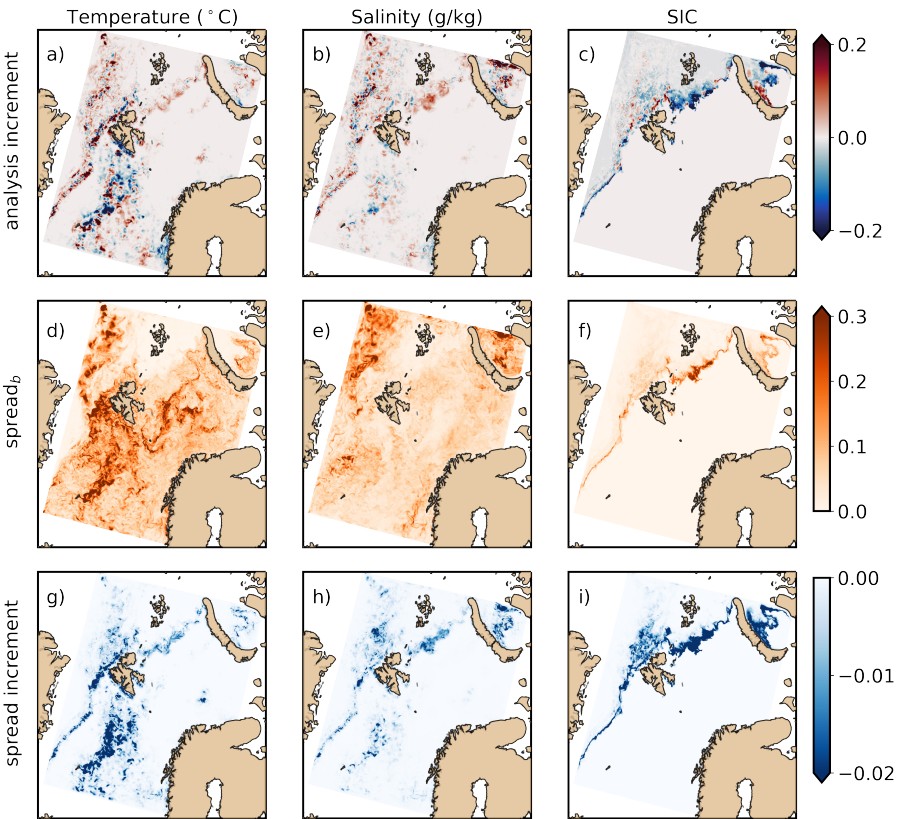

**Figure 15.** Analysis increments of (a) SST, (b) surface salinity, and (c) SIC for an assimilation cycle at 2022-12-15, as calculated by the EnKF. Ensemble spread of the model background state for (d) SST, (e) surface salinity, and (f) SIC. Spread increments for (g) SST, (h) surface salinity, and (i) SIC. Analysis (spread) increments correspond to the analysis minus the background state (spread).

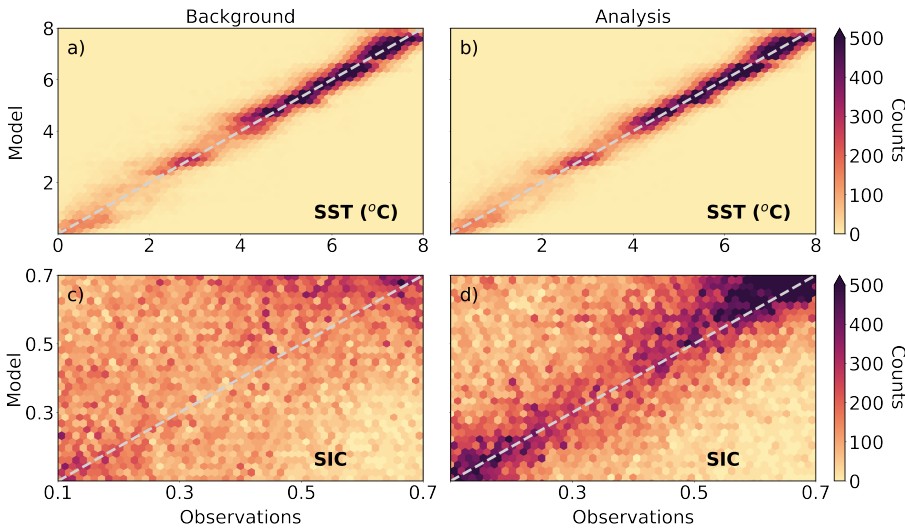

**Figure 16.** Binned scatter plot of model versus observation values for (a, b) SST and (c, d) SIC. Panels (a, c) show a comparison for the background model state, and (b, d) for the analysis on 2022-12-15, i.e. the same DA cycle as in Fig. 15, based on the observations used in Figs. 2 and 11.

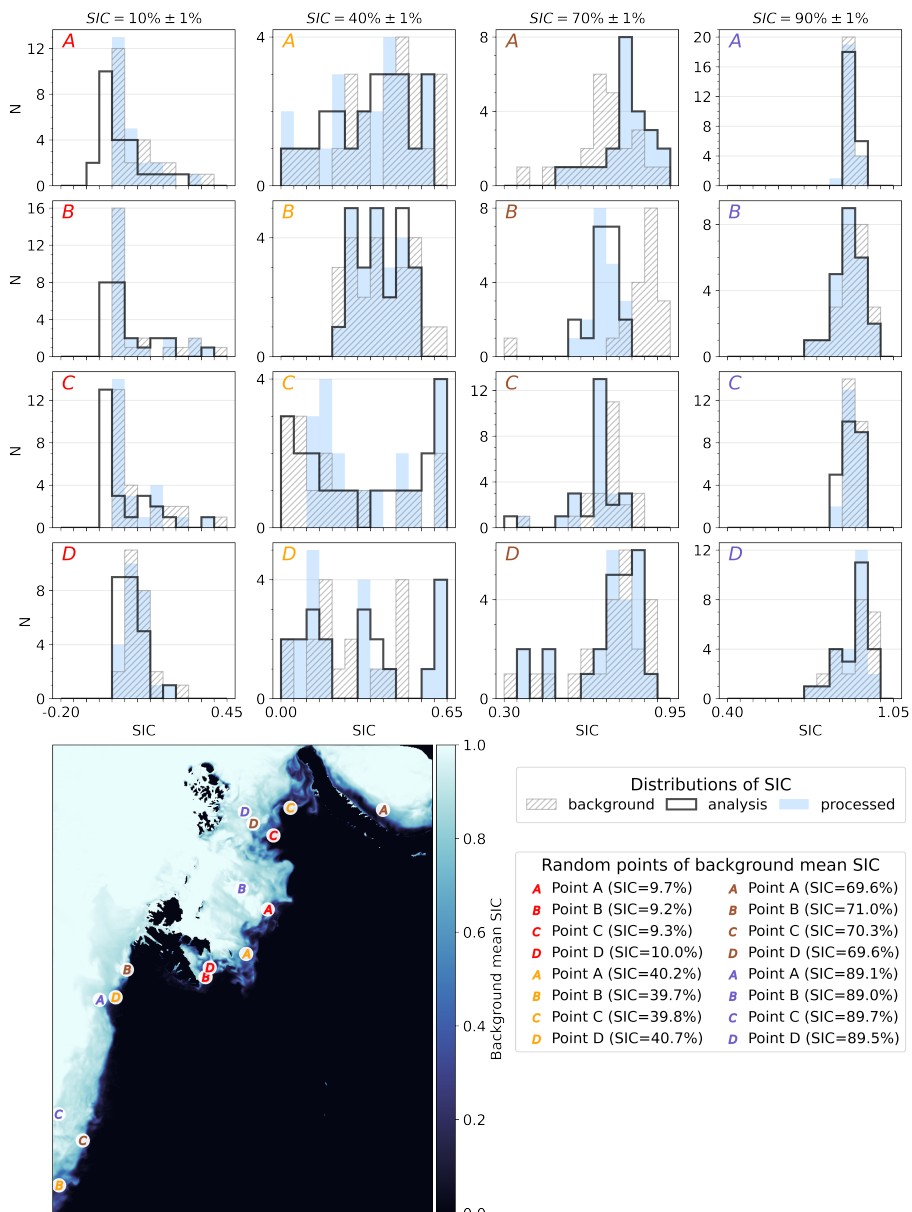

**Figure 17.** Distribution of SIC across the ensemble at random points in the extended marginal ice zone. Distributions are shown for each point marked in the map panel, representing various ice conditions. The point locations are chosen from a set of defined SIC ranges, i.e., $(\{10, 40, 70, 90\} \pm 1)\%$ SIC. The histograms show SIC distributions of the background model state (background), for the analysis computed by the EnKF (analysis), and for the post-processed SIC fields that is used to re-initialize the model (processed). The post-processed SIC may differ from the EnKF analysis if any of the SIC classes contain values outside the range of $0 < SIC < 100\%$. All histograms have bin widths of 5% SIC, and the vertical axes shows the number of ensemble members, $N$, within each bin.