# Peer review of "Barents-2.5km v2.0: An operational data-assimilative coupled ocean and sea ice ensemble prediction model for the Barents Sea and Svalbard"

_Geoscientific Model Development, 2023_

## Referee Comment (RC2)

Review for "Barents-2.5km v2.0: An operational data-assimilative coupled ocean and sea ice ensemble prediction model for the Barents Sea and Svalbard"

**Recommendations: Major Revision**

**General Comments and summary**

This paper presents the version 2.0 of the operational ocean and sea-ice forecast Barents-2.5km model. This version includes an ensemble prediction system (EPS) with an off-line ensemble-based data assimilation (DA) component. The system routinely assimilates sea ice concentration (SIC), sea surface temperature (SST), and in-situ hydrography observations. With the DA component, the Barents-2.5 km model shows better SST forecast skills, e.g., improvement over the persistence forecast during spring and summer. Although the predictive skill for SIC is not as high, the model still displays skillful performance with DA, as it reduces the model drift away from the truth state. Furthermore, the EPS also provides the uncertainty estimate of the model state. The ensemble spread for SST is generally reasonable, although it may miss some extreme values, whereas the ensemble spread for SIC is too small. Overall, the Barents-2.5km model with its EPS and DA component is a valuable tool for forecasting ocean and sea-ice conditions.

After carefully reviewing the paper, I am impressed with the technical details presented. The work is certainly worthy of publication in GMD. However, I have some concerns with the DA part. In particular, some of the context regarding the ensemble Kalman filter (section 3.1) appears to be incorrect, and in my opinion, some important DA details seem to be missing. Although the paper does include some analysis on the performance of DA, the issue of non-Gaussianity, which can be especially important for sea ice observation, is not much addressed or discussed. While I appreciate the manuscript's primary focus on documenting and demonstrating technical expertise, it's equally important to ensure that the context of the DA component is accurately and completely presented. Therefore, I recommend a major revision of the DA section before publication in GMD.

**Specific line-by-line comments**

- Line 178

The citation here can be a little misleading, as the EnKF in (Evensen 1994; Burgers et al. 1998) are not usually referred to as the deterministic version of EnKF. I suggest putting the citation (Sakov and Oke 2008) here, and move the citations (Evensen 1994; Burgers et al. 1998) to line 180.

For the EnKF reference, in addition to (Evensen 1994; Burgers et al. 1998), I recommend that also include another reference (Houtekamer and Mitchell, 1998)

Houtekamer, P.L. and Mitchell, H.L. (1998) Data assimilation using an ensemble Kalman filter technique. Monthly Weather Review, 126, 796–811.

- Line 183-184

Although it is tangent to the main thread of the paragraph here, van Leeuwen (2020) notes that in the original stochastic EnKF, the perturbations should be added to the ensemble equivalence of the observation $H(x)$ instead of the observation $y$. This distinction becomes significant when the observation error is non-symmetric (e.g., skewed), which can have important implications for, e.g., bounded observations.

van Leeuwen, PJ. (2020) A consistent interpretation of the stochastic version of the Ensemble Kalman Filter. QJR Meteorol Soc., 146: 2815– 2825. https://doi.org/10.1002/qj.3819

- Line 185-190

Equation (2) is the stochastic version of EnKF, while this paper uses the deterministic version of EnKF. Using Equation (2) here can be confusing to the readers. Therefore, I suggest, e.g., replacing Equation (2) with the deterministic transform equation in Sakov and Oke (2008) and replacing this paragraph with a new one (or add a new one) for the deterministic EnKF in Sakov and Oke (2008).

- Line 197-198

Is this a reasonable assumption for the observations assimilated in this work? This assumption will introduce larger representation error to the observations that are taken at time points more distant from the analysis time. Although this issue is discussed in Section 6.4, I suggest adding one or two sentences commenting on this assumption here.

- Line 199-202

I suggest extending this paragraph by adding more DA details. Specifically,

(1) Including some details about what the "spread reduction factor" and the "global moderation factor" mean, and how they work.

(2) It seems that only horizontal localization is applied. Do the sea surface observations have impact on the state variables in the ocean (e.g., the ocean current at 30-m deep)?

(3) I suggest providing more information on how the observation errors are moderated, since it is one of the key part in DA system. I wonder if the adaptive tuning of the observation error could somehow partially compensate the issue of the time-dependent representation error for the observations.

(4) Are the observations the same type of variables as the model states in ROMS/CICE? i.e., are the observation operators just interpolating the model state to the observation location?

- Line 291

Why are the SST validated for these regions (defined in Fig. 6) separately? Are there any important implications from Fig. 7? I suggest adding one or two sentences briefly discussing the results from Fig. 7.

- Line 304

What kind of failures in DA are specifically referred to here?

- Line 347-349

I suggest including the reference(s) for the rank histogram, e.g., (Hamill 2001).

Hamill, T.M. (2001) Interpretation of rank histograms for verifying ensemble forecasts. Monthly Weather Review, 129, 550–560. https://doi.org/10.1175/15200493(2001)129 2.0.CO;2.

- Lines 353-355

I suggest incorporating more descriptions on how the reliability diagram is generated, and the way to interpret the reliability diagram.

- Line 366-367

Similar to the previous comment, e.g., why does the reversed S-shape indicate low ensemble spread?

- Line 374-396 (general comment for section 5.4)

It is interesting to also discuss the analysis increment for the unobserved variables, e.g., ocean current.

- Line 384

I suggest being more specific here. For example, revise "The correlation for SST is…" to "the correlation between … and … is"

- Line 481-483

While I do not insist on conducting more DA experiments to address the following issues in this paper, it would be helpful to include some of the discussions regarding the following questions:

(1) Could a larger inflation factor lead to a better DA and forecast performance?
(2) Exploring whether techniques to address insufficient ensemble spread, e.g., see the list below, can be effective would be an interesting experiment in the future work as well.

Zhang F. Q., Snyder C., Sun J. Z. (2004), Impacts of initial estimate and observation availability on convective-scale data assimilation with an ensemble Kalman filter. Mon. Weather Rev. 132: 1238–1253.

Anderson, J. L. (2007), An adaptive covariance inflation error correction algorithm for ensemble filters. Tellus A, 59: 210-224. https://doi.org/10.1111/j.1600-0870.2006.00216.x

Anderson, J. L. (2009), Spatially and temporally varying adaptive covariance inflation for ensemble filters. Tellus A, 61: 72-83. https://doi.org/10.1111/j.1600-0870.2008.00361.x

Whitaker J. S., and Hamill T.M. (2012), Evaluating methods to account for system errors in ensemble data assimilation. Mon. Weather Rev. 140:3078–3089.

Ying, Y., and Zhang, F. (2015). An adaptive covariance relaxation method for ensemble data assimilation. Quarterly Journal of the Royal Meteorological Society, 141(692), 2898-2906.

El Gharamti, M. (2018). Enhanced adaptive inflation algorithm for ensemble

filters. Monthly Weather Review, 146(2), 623-640.

(3) Since SIC is a bounded variable, assuming Gaussian error for SIC is inappropriate especially when SIC value is close to the boundary (i.e., zero or one). This non-Gaussianity can make Gaussian DA method, like EnKF, sub-optimal. It would be helpful to check the ensemble distribution of SIC at a single location before and after DA in a single DA cycle, (1) when SIC observation is close to the boundary, e.g., SIC = 0 or 1, (2) when SIC observation is away from the boundary, e.g., SIC = 0.5. Using some non-Gaussian DA techniques (e.g., Bishop 2016; Poterjoy 2016; Hu and van Leeuwen 2021; Anderson 2022; Chan et al. 2023, etc) can alleviate this problem and may improve the assimilation and the forecast of SIC.

Bishop, C.H. (2016), The GIGG-EnKF: ensemble Kalman filtering for highly skewed non-negative uncertainty distributions. Q.J.R. Meteorol. Soc., 142: 1395-1412. https://doi.org/10.1002/qj.2742

Poterjoy, J. (2016). A localized particle filter for high-dimensional nonlinear systems. Monthly Weather Review, 144(1), 59-76. https://doi.org/10.1175/MWR-D-15-0163.1

Hu, C. C., & Van Leeuwen, P. J. (2021). A particle flow filter for high-dimensional system applications. Quarterly Journal of the Royal Meteorological Society, 147(737), 2352-2374. https://doi.org/10.1002/qj.4028

Anderson, J. L. (2022). A Quantile-Conserving Ensemble Filter Framework. Part I: Updating an Observed Variable, Monthly Weather Review, 150(5), 1061-1074. https://doi.org/10.1175/MWR-D-21-0229.1

Chan, M.-Y., Chen, X., & Anderson, J. L. (2023). The potential benefits of handling mixture statistics via a bi-Gaussian EnKF: Tests with all-sky satellite infrared radiances. Journal of Advances in Modeling Earth Systems, 15, e2022MS003357. https://doi.org/10.1029/2022MS003357

- Figure 9
In the caption, I suggest also adding the meaning of the other lines (blue, blue dashed, orange, orange dashed).

- Figure 11

(1) Why there are more than 4 lines in each panel (there are only 4 in the legend)?

(2) What are the values in x-axis? Why are they not chosen uniformly between [0,1]?

(3) line 2 in the caption: … stating hos well … -> stating how well

- Figure 13

I recommend adding the unit to the variables in the figure.

---

## Author Comment (AC1)

**Replies to Reviewer #1: Review of "Barents-2.5km v2.0: An operational data-assimilative coupled ocean and sea ice ensemble prediction model for the Barents Sea and Svalbard"**

*General Comments*

*This paper presents the development methods and validation of the Barents-2.5km v2.0 forecast model. The new version of the Barents-2.5km model has been upgraded to include a 24 member Ensemble Prediction System. The data assimilation has also been upgraded from only sea ice concentration to additionally including sea surface temperature and hydrographic profiles.*

*Overall the paper shows the forecast system has promise as a useful tool for various applications that require predictions of ocean variables in the Barents, Norwegian and Greenland Sea. This is particularly important for the marginal ice zone areas and the paper shows the challenge and skill required for accuracy. Some of the description of the model forecast skill is hard to follow. Analysis and validation of currents are presented in a separate paper which means there is not a complete picture here. There is also no comparison to v1.0 of the Barents-2.5km model. There are a few points that need attention that I detail below.*

Thank you for your feedback on our manuscript, and a list of specific comments to address. In order to present a fuller assessment of the general model performance for the most relevant state variables, we aim to present the statistical properties of ocean currents and ice drift vectors as compared to observations, and put some focus on validation by non-observed variables.

*Specific Comments*

*Line 7-9: Currents are only briefly mentioned in the paper and seems to be a larger part of another paper. The abstract should be used to discuss what is in the paper.*

Parts of the ocean current analysis will be included in the revised manuscript, focusing on the statistical skill of surface currents and ice drift from the reference member. A more comprehensive analysis of surface current predictability and from the same model will be presented in another document, but to avoid confusion we will remove the reference to this work from our abstract.

*Line 108-110: Have you considered using the elastic-plastic anisotropic (EAP) rheology scheme in CICE (Heorton et al., 2018)? Why was EVP chosen?*

The EVP rheology was chosen in this model setup because it is computationally faster, which has a bearing on our capability to run multiple ensemble members in a forecast mode beside ROMS. A comment on this will be included in the manuscript.

*Line 291: You should mention the SST RMSE July peak.*

This feature will be mentioned along with the other peaks in the revised manuscript.

*Lines 293-296: The model is compared against the same observations that are entrained through the data assimilation system. I understand that the observations are considered independent because they are compared with a previous forecast. This has value in quantifying the quality of the prediction for assimilated variables. However, I think the quality of the paper could be improved if you make comparison with un-assimilated data, perhaps sea surface height which could give an indication of current quality.*

We agree that using observation types that were not assimilated at all would allow a more complete assessment of the model performance. We think that both upper ocean current velocities (from HF radar) and low-frequency ice drift vectors (from passive microwave imagery) could fill this gap. For an independent validation of model hydrography, we are able to use float observations during model periods when the EnKF was not successful, hence these data were not assimilated and provide a means for validation. Such results are envisioned for the revised manuscript.

*Line 300-305: Please add the SIC ice charts dataset to the methods and reference.*

The subjective sea ice concentration charts are another pivot of independent observations used for validation. We will include documentation on this data set in a new method section and add a remark that these data are not assimilated in the model.

*Line 312-314: Sea ice concentration tends to increase rapidly when there is no data for assimilation. It seems to be reliant of the SIC assimilation data. I see you've made a point about why this happens in Line 422 but I think it would be better to make that point here or rephrase the bit around Line 422.*

We will add this comment in Sec. 5.2 as suggested.

*Line 330-334: Why does the model have greater RMSE for SST up to 12h lead time than the persistence, deterministic or trend with observations?*

By definition, the persistence forecast delivers zero RMSE at lead time zero (e.g. the persistence reference forecast says that SST of the future is the same as now). The trend forecast same as the persistence plus the time-change from the model, hence it is also zero at lead-time zero. The deterministic is the reference member from the Barents-EPS, it may have an RMSE smaller or larger than the ensemble mean. We clarify these points in the revised manuscript.

*Line 347-358: Describe the validation/analysis technique in the methods. Same goes for other validation methods. What do the bins represent?*

A new method section that describes our validation methods will be added in the revised manuscript.

*Line 374-377: What are analysis increments? This needs to be clearer.*
*Line 380: How is the spread calculated? Is it the standard deviation of the ensemble?*
*Line 392-393: What are spread increments? It also does not say what they are in Figure 12 caption.*
*Line 442-446: I do not understand Figures 10 and 11 enough. Why is the spread in SST satisfactory? Why is the spread in SIC too low?*

Analysis increments are the difference between the model analysis (i.e. the model state after assimilation) and the model background (i.e. the model state before data assimilation) at the same time, i.e. the analysis time step. Similarly, spread increments are the difference between the analysis ensemble spread (after assimilation) and the background ensemble spread (before assimilation). We will include all these definitions in the revised text.

The ensemble spread for each state variable is the standard deviation between ensemble members. The spread is typically reduced (i.e. the spread increment is negative) during the assimilation process, as all members are brought closer towards the observations.

Figs. 10,11 and 12 will be improved for clarity and layout, and we will put more emphasis on the text discussing these figures, where the quality of the ensemble spread is assessed. For the rank histogram (Fig.10), an ideal ensemble spread would yield a flat uniform distribution, i.e. all members have an equal probability of realization. Underdispersive ensembles are identified by accumulation of observations in the lowest and highest ranks, meaning that the model values rarely yield as high or low values as seen in observations. We extend the discussion in the revised manuscript and provide references on ensemble modeling techniques. In the reliability diagram (Fig. 11), an ideal spread would be given by a straight diagonal line, indicating that modeled probabilities of exceeding threshold match observed frequencies of exceeding these threshold. Deviations from the diagonal show that the EPS tends to over- or underestimate probabilities to exceed certain ice concentration thresholds. In our case, the EPS underestimates probabilities for occurrence of open water and low ice concentrations.

*Line 471-472: How does the Barents-2.5km v2.0 of the model compare against original Barents-2.5km? Can you quantify the improvement in accuracy that users can expect?*

We have no period that is suitable for direct comparison as the v1.0 model was prone to failures during the transition to the new model system. However, we will provide a discussion on the changes in model performance that the user may expect:

- The v1 model applied a strong constraint to observed ice concentration using a nudging scheme, and hence matched AMSR maps closely at the analysis time step. Because ocean temperatures were not modified during the analysis, the ice concentration forecast deteriorated rapidly.

- No ocean temperature was assimilated in the v1 model setup. As a consequence, ocean temperature and salinity drifted away from the observed ocean state. Using a nudging towards the parent model state, the Barents v.1 reflected the hydrography of that model (Topaz).

*Table 1: This could be quoted in the text instead of a table.*

 We accept this suggestion.

The Technical corrections will be implemented.

*Figure 12: Please improve the caption with more detail. What are the black regions in g and h?.*

Fig. 12 will be updated. The black regions in g and h have been removed to make figures clearer. These dark areas corresponded to zones of zero spread increments, thus where the spread was not modified after assimilation. These zones correspond now to white areas with zero as colorbar value.

---

## Author Comment (AC2)

**Response to Reviewer #2: Review for "Barents-2.5km v2.0: An operational data-assimilative coupled ocean and sea ice ensemble prediction model for the Barents Sea and Svalbard"**

*Recommendations: Major Revision*

*General Comments and summary*

*This paper presents the version 2.0 of the operational ocean and sea-ice forecast Barents-2.5km model. This version includes an ensemble prediction system (EPS) with an off-line ensemble-based data assimilation (DA) component. The system routinely assimilates sea ice concentration (SIC), sea surface temperature (SST), and in-situ hydrography observations. With the DA component, the Barents-2.5 km model shows better SST forecast skills, e.g., improvement over the persistence forecast during spring and summer. Although the predictive skill for SIC is not as high, the model still displays skillful performance with DA, as it reduces the model drift away from the truth state. Furthermore, the EPS also provides the uncertainty estimate of the model state. The ensemble spread for SST is generally reasonable, although it may miss some extreme values, whereas the ensemble spread for SIC is too small. Overall, the Barents-2.5km model with its EPS and DA component is a valuable tool for forecasting ocean and sea-ice conditions.*

*After carefully reviewing the paper, I am impressed with the technical details presented. The work is certainly worthy of publication in GMD. However, I have some concerns with the DA part. In particular, some of the context regarding the ensemble Kalman filter (section 3.1) appears to be incorrect, and in my opinion, some important DA details seem to be missing. Although the paper does include some analysis on the performance of DA, the issue of non-Gaussianity, which can be especially important for sea ice observation, is not much addressed or discussed. While I appreciate the manuscript's primary focus on documenting and demonstrating technical expertise, it's equally important to ensure that the context of the DA component is accurately and completely presented. Therefore, I recommend a major revision of the DA section before publication in GMD.Specific line-by-line comments*

Thank you for the appreciation of our work on the model, and the feedback on our manuscript. We would like to improve our manuscript towards a better description of the applied DA methods, making the corrections of errors noted by the reviewer. We also aim to add details about the assimilation of sea ice concentrations using the deterministic EnKF. In particular, we want to present an analysis of the local gaussianity (or lack thereof) in sea ice concentrations. We agree that this aspect is important, as we do see limitations of our model's capabilities towards constraining sea ice concentrations in an efficient way and future model development has to address this shortcoming.

We also appreciate the reference suggestions provided by the reviewer for several aspects of the EnKF method and sea ice assimilation.

*- Line 178 The citation here can be a little misleading, as the EnKF in (Evensen 1994; Burgers et al. 1998) are not usually referred to as the deterministic version of EnKF. I*

*suggest putting the citation (Sakov and Oke 2008) here, and move the citations (Evensen 1994; Burgers et al. 1998) to line 180. For the EnKF reference, in addition to (Evensen 1994; Burgers et al. 1998), I recommend that also include another reference (Houtekamer and Mitchell, 1998)*

This paragraph will be adapted to include the suggested citations and make a clearer difference between the EnKF and DEnKF, mainly regarding formulas and citations. In particular, the approximated ensemble transform matrix used in the DEnKF will be specified in the text.

*- Line 183-184: Although it is tangent to the main thread of the paragraph here, van Leeuwen (2020) notes that in the original stochastic EnKF, the perturbations should be added to the ensemble equivalence of the observation H(x) instead of the observation y. This distinction becomes significant when the observation error is non-symmetric (e.g., skewed), which can have important implications for, e.g., bounded observations. van Leeuwen, PJ. (2020) A consistent interpretation of the stochastic version of the Ensemble Kalman Filter. QJR Meteorol Soc., 146: 2815– 2825. https://doi.org/10.1002/qj.3819*

This is an interesting detail on how ensemble perturbations are computed. We will add a remark in section 3.1 of the revised manuscript.

*- Line 185-190: Equation (2) is the stochastic version of EnKF, while this paper uses the deterministic version of EnKF. Using Equation (2) here can be confusing to the readers.* **Therefore, I suggest, e.g., replacing Equation (2) with the deterministic transform equation in Sakov and Oke (2008) and replacing this paragraph with a new one (or add a new one) for the deterministic EnKF in Sakov and Oke (2008).**

Our revised manuscript will contain a separate paragraph on the deterministic EnKF (DEnKF);  in particular, the approximated ensemble transform matrix used in the DEnKF is specified in the text together with the reference to Sakov and Oke (2008). Prior to this paragraph, an introductory paragraph about the EnKF is kept with the corresponding general EnKF equations (Eq. 2 and Eq. 3) and references. The suggested reference of Houtekamer and Mitchell (1998) will be added as an EnKF reference in the text.

*- Line 197-198 Is this a reasonable assumption for the observations assimilated in this work? This assumption will introduce larger representation error to the observations that are taken at time points more distant from the analysis time. Although this issue is discussed in Section 6.4, I suggest adding one or two sentences commenting on this assumption here.*

Synchronous assimilation of observed variables is clearly a simplification, and currently we are working on experiments to study the benefits of asynchronous assimilation. We will add more details on the implications of this assumption  in the revised manuscript. In essence, we argue that the synchronous DA may be sufficient to avoid drift of the model state. Most important shortcomings are in situations with diurnal variation as for  summer time SSTs during low winds.

*Line 199-202 I suggest extending this paragraph by adding more DA details. Specifically, (1) Including some details about what the "spread reduction factor" and the "global moderation factor" mean, and how they work.*

We will add more explanations of the parameters used to configure the EnKF.

*(2) It seems that only horizontal localization is applied. Do the sea surface observations have impact on the state variables in the ocean (e.g., the ocean current at 30-m deep)?*

It is true that only horizontal localisation is applied for the EnKF in our model. With further investigation of model results following this remark, we note in fact that in some circumstances the lack of vertical localisation combined with relatively low number of ensemble members can lead to artifacts in the model analysis at depth, possibly resulting from spurious correlations in the ensemble. We will discuss this point in a revised manuscript, including possible ways forward to address this issue.

*I suggest providing more information on how the observation errors are moderated, since it is one of the key part in DA system. I wonder if the adaptive tuning of the observation error could somehow partially compensate the issue of the time-dependent representation error for the observations.*

In `enkf-c`, there is a moderation factor (K factor, see equation below) used to moderate the observation impact by smoothly increasing the observation error in function of the innovation magnitude. For large innovations, the K-factor plays an important role in the increase of the observation error; for weak innovations, the K-factor does not play a big role and observations errors are mostly kept unchanged. A K-factor equal or lower than 2 is recommended; if the K-factor is set too high, then for large innovations we would lose the moderation term to decrease the observation impact and we want to avoid that.

$$\sigma^2_{obs} \leftarrow [(\sigma^2_b + \sigma2obs)2 + \sigma2b \ d2/K2]^{\frac{1}{2}} - \sigma2b$$

*Are the observations the same type of variables as the model states in ROMS/CICE? i.e., are the observation operators just interpolating the model state to the observation location?*

At present, our model system directly maps the model state variables to observed parameters. We think that this is a fair assumption for in-situ hydrography and sea ice concentration from passive microwave imagery. For sea surface temperature as observed by infrared and visible band imagery, we acknowledge a mismatch during skin temperature and upper model layer temperature. Due to the far north location of the model domain, this mismatch is limited but visible during spring and summer time. We are currently working on the implementation of an observation operator for the skin temperature. During most of the time however, our model domain is either well-mixed due to wind forcing or experiencing relatively low levels of radiative forcing that drive the differences between upper layer and skin temperatures.

*- Line 291 Why are the SST validated for these regions (defined in Fig. 6) separately? Are there any important implications from Fig. 7? I suggest adding one or two sentences briefly discussing the results from Fig. 7.*

The used areas for SST validation characterize areas with distinct hydrography, e.g. water masses, water depth and sea ice climatology. The same areas are used in a validation of the Arctic CMEMS product. We will add this explanation in the text.

*- Line 304 What kind of failures in DA are specifically referred to here?*

Most common have been IT-related issues (e.g. lack of sufficient memory allocated to the EnKF task), and lack of exceptions dealt with in the queuing system (e.g. failures of the

scheduling system and human mistakes during code updates) . At few instances, the EnKF-c software failed due to observations or model states that resulted in the software exiting.

*- Line 347-349 I suggest including the reference(s) for the rank histogram, e.g., (Hamill 2001). Hamill, T.M. (2001) Interpretation of rank histograms for verifying ensemble forecasts.*

*- Lines 353-355: I suggest incorporating more descriptions on how the reliability diagram is generated, and the way to interpret the reliability diagram.*

*- Line 366-367: Similar to the previous comment, e.g., why does the reversed S-shape indicate low ensemble spread?- Line 374-396 (general comment for section 5.4)*

We will add these references and a more detailed explanation of the ensemble verification methods.

*It is interesting to also discuss the analysis increment for the unobserved variables, e.g., ocean current.*

The increments for unobserved variables, such as current, will be discussed in a revised manuscript given an example similar to the analysis increments for observed variables in Fig. 12.

*- Line 384 I suggest being more specific here. For example, revise "The correlation for SST is..." to "the correlation between ... and ... is"*

We will rephrase this part to be more specific.

*- Line 481-483 While I do not insist on conducting more DA experiments to address the following issues in this paper, it would be helpful to include some of the discussions regarding the following questions: (1) Could a larger inflation factor lead to a better DA and forecast performance? (2) Exploring whether techniques to address insufficient ensemble spread, e.g., see the list below, can be effective would be an interesting experiment in the future work as well.*

During DA experiments we have seen that larger inflation factors for sea ice variables do improve the validation metrics for sea ice concentration. However, as a result of the inflation we also get artifacts in other variables such as sea ice thickness and current velocities. We therefore keep inflation to relatively low levels.

Addressing insufficient ensemble spread in sea ice concentrations, we do not have a final conclusion how this could be improved but are currently experimenting with various configuration setups for the EnKF. We would like to mention some of these strategies in a revised paper, but are not yet ready to provide results. In particular, we have recently noticed that thinning of sea ice concentration observations in the DA step may improve ensemble spread.

*(3) Since SIC is a bounded variable, assuming Gaussian error for SIC is inappropriate especially when SIC value is close to the boundary (i.e., zero or one). This non- Gaussianity can make Gaussian DA method, like EnKF, sub-optimal. It would be helpful to check the ensemble distribution of SIC at a single location before and after DA in a single DA cycle, (1) when SIC observation is close to the boundary, e.g., SIC = 0 or 1, (2) when SIC observation is away from the boundary, e.g., SIC = 0.5. Using some non-Gaussian DA techniques (e.g.,*

*Bishop 2016; Poterjoy 2016; Hu and van Leeuwen 2021; Anderson 2022; Chan et al. 2023, etc) can alleviate this problem and may improve the assimilation and the forecast of SIC.*

We follow your suggestion to include a brief analysis of the local distribution of SIC in the ensemble, for the background and analysis. An example of SIC distribution will be provided in a new Figure. We see gaussian-like distributions for intermediate ice concentrations in the marginal ice zone, and certain spread at low ice concentrations. At SIC close to 1, gaussianity of SIC is often broken and we expect deficiencies in the DA method in these areas. It is also worth mentioning that the EnKF may provide increments with SIC > 1 or SIC < 0 , and in those cases we limit the increment such that 0 < SIC < 1. We include a remark on methods to deal with non-Gaussian variables (suggested references) and these are considered in our future model development.

---

## Author Response (AR1)

**Summary of major changes in response to reviewer 1&2 comments on the manuscript**

- The revised manuscript includes more details on the EnKF DA method applied in the model (Section 3), referring first to the EnKF method in general, followed by the DEnKF method applied in this work. More details on the configuration of the EnKF is provided, including the moderation of observation error.
- A new section (Section 4) focuses on the observations used for both DA and validation.
- Model results are now also validated against independent (i.e. non-assimilated) observations of in-situ hydrography, ice drift, surface currents, and ice charts.
- A more in-depth description of validation methods is provided in a new section (Section 6).
- More details on interpretation of rank histogram and reliability diagram to assess ensemble spread is given in Section 6.
- We included a more detailed discussion of methods to assimilate sea ice concentration using the EnKF DA scheme, including an analysis of sea ice concentration distribution in the ensemble in Section 8.2.
- Spelling and grammar issues were corrected throughout the manuscript, and in some cases sentences were simplified.

**Responses to reviewer 1 specific comments**

*Line 7-9: Currents are only briefly mentioned in the paper and seems to be a larger part of another paper. The abstract should be used to discuss what is in the paper.*

We have included a new section (Section 7.1 Validation against independent observations) in the revised manuscript, which focuses on (i) the statistical skill of surface currents (line 402-408) and (ii) ice drift (line 409-413). A more comprehensive analysis of surface current predictability from the same model is presented in another document. To avoid confusion, we have removed the reference to this work from the abstract.

*Line 108-110: Have you considered using the elastic-plastic anisotropic (EAP) rheology scheme in CICE (Heorton et al., 2018)? Why was EVP chosen?*

The EVP rheology was chosen in this model setup because it is computationally faster, which has a bearing on our capability to run multiple ensemble members in a forecast mode beside ROMS. A comment on this has been included in the manuscript (line 107-109): *"In this work, CICE is configured to use elastic-viscous-plastic (EVP) rheology (Hunke and Dukowicz, 1997), which is a compromise for computational cost compared to the elastic-anisotropic rheology (e.g. Heorton et al., 2018)."*

*Line 291: You should mention the SST RMSE July peak.*

This feature has been addressed in the revised manuscript in Section 7.2 (line 417-418): *"Model errors in SST peak during summer months, which is the time period with the most rapid changes in SST and in skin temperature driven by short-wave radiation."*

We also comment on the difficulty to resolve summer SST during strong solar radiation in Section 8.1 (line 541-543): *"Due to the stronger solar forcing, SST in summer time changes more rapidly than during the winter season. Barents-2.5 is forced by hourly fields from MET-AA, giving SST forecasts an advantage over the persistence forecast during spring and summer (Fig. 12)."*

*Lines 293-296: The model is compared against the same observations that are entrained through the data assimilation system. I understand that the observations are considered independent because they are compared with a previous forecast. This has value in quantifying the quality of the prediction for assimilated variables. However, I think the quality of the paper could be improved if you make comparison with un-assimilated data, perhaps sea surface height which could give an indication of current quality.*

We agree that using observation types that were not assimilated at all allows a more complete assessment of the model performance. Therefore, both upper ocean current velocities (from HF radar) and low-frequency ice drift vectors (from passive microwave imagery) were used for an independent validation. For the independent validation of model hydrography, we used float observations during model periods when the EnKF was not successful, hence these data were not assimilated and provide a means for validation. These results are included in the revised manuscript in Section 7.1. All used observation types are described in Section 4.

*Line 300-305: Please add the SIC ice charts dataset to the methods and reference.*

The subjective sea ice concentration charts are another pivot of independent observations used for validation. The data set is described in Section 4.4 of the revised manuscript. In Section 6.2, we describe how the subjective ice charts are used for validation (line 358-364). The ice chart data is not assimilated in the model.

*Line 312-314: Sea ice concentration tends to increase rapidly when there is no data for assimilation. It seems to be reliant of the SIC assimilation data. I see you've made a point about why this happens in Line 422 but I think it would be better to make that point here or rephrase the bit around Line 422.*

We have added this remark in Section 7.2: *"Likewise, the model performance deteriorates within a few days when no DA is applied for subsequent cycles, as seen for two periods during March and May 2022 in Fig. 8. A positive SIC bias is present in the model during periods without DA, possibly as a result of too fast ice drift velocities resulting in the ice*

*cover to extend into open water areas.",* and address this again in the discussion (Section 8.1, line 530-535).

*Line 330-334: Why does the model have greater RMSE for SST up to 12h lead time than the persistence, deterministic or trend with observations?*

By definition, the persistence forecast delivers zero RMSE at lead time zero (i.e. the persistence reference forecast says that SST of the future is the same as now). The trend forecast is the same as persistence plus the time-change from the model, hence it is also zero at lead-time zero. By deterministic, we mean the reference member from the Barents-EPS, which may have an RMSE smaller or larger than the ensemble mean. We have explained these methods in the revised manuscript in Section 6.1 (line 342-348) and provide an interpretation in Section 7.3 (line 436-448).

*Line 347-358: Describe the validation/analysis technique in the methods. Same goes for other validation methods. What do the bins represent?*

A new section has been added for the description of our methods, Section 6 Validation methods. Both the rank histogram and reliability diagram are explained here, and we include references for these methods (Hamil 2001, Bröcker and Smith, 2007). The figure caption has been revised, and now includes more a specific description of the generation of the diagram, e.g., for the bins; *"The forecast probabilities calculated for each interval are grouped into evenly spaced bins from 0.0 to 1.0 on the x-axis. [...]"* and they are now shown explicitly in the plot as well.

*Line 374-377: What are analysis increments? This needs to be clearer.*
*Line 380: How is the spread calculated? Is it the standard deviation of the ensemble?*
*Line 392-393: What are spread increments? It also does not say what they are in Figure 12 caption.*
*Line 442-446: I do not understand Figures 10 and 11 enough. Why is the spread in SST satisfactory? Why is the spread in SIC too low?*

A clarification of the analysis increment has been added in Section 7.5 (line 475-476):
*"For each state variable, the analysis increment is the difference between the new model analysis provided by the EnKF and the model background, i.e. the forecast of the previous cycle."*
A clarification of the calculation of spread has been added in Section 7.4 (line 490):
*"Ensemble spread - which is the standard deviation computed from the ensemble of each state variable [...]"*
A clarification of the spread increments has been added (line 492):
*"Spread increments, i.e. the difference between ensemble spread before and after DA [...]"*

The figure caption of Fig. 15 (previously Fig. 12) has been updated to include a reminder as well: *"Analysis (spread) increments correspond to the analysis minus the background state (spread)."*

Figs. 13 (rank histogram, previously Fig. 10) and 14 (reliability diagram, previously Fig. 11) have been improved for clarity and layout. Their figure captions now include a much more descriptive presentation of the content and generation of the diagrams. In addition, we have included a thorough description of the methods to generate these plots in Sections 6.1 and 6.2. The method section now also describes the characteristics we expect to see for different versions of ensemble spread, both in the rank histogram and reliability diagram. In Section 7.4 Ensemble spread, these two figures are presented and we have made sure to remind the reader of the features again, e.g., "*We see a fair ensemble spread for this variable across the center part (i.e. a flat distribution in the rank histogram) [...].*"

*Line 471-472: How does the Barents-2.5km v2.0 of the model compare against original Barents-2.5km? Can you quantify the improvement in accuracy that users can expect?*

We have no period that is suitable for direct comparison as the v1.0 model was prone to failures during the transition to the new model system. However, we provide a brief discussion in Section 8.4 (line 610-614) on the changes in model performance that the user may expect:

- The v1 model applied a strong constraint to observed ice concentration using a nudging scheme, and hence matched AMSR maps closely at the analysis time step. Because ocean temperatures were not modified during the analysis, the ice concentration forecast deteriorated rapidly.
- No ocean temperature was assimilated in the v1 model setup. As a consequence, ocean temperature and salinity drifted away from the observed ocean state. Using nudging towards the parent model state, the Barents v.1 reflected the hydrography of that model (TOPAZ4).

*Table 1: This could be quoted in the text instead of a table.*

We have included the tabulated values, previously in Tab. 1, into the text in Section 2.1.1 Discretization and advection schemes (line 75).

*Figure 12: Please improve the caption with more detail. What are the black regions in g and h?*

The figure is updated in the revised manuscript (now Fig. 15). The black regions in g and h have been removed to make the figures clearer. These dark areas corresponded to zones of zero spread increments, thus where the spread was not modified after assimilation. These zones correspond now to white areas with zero as colorbar value.

**Responses to reviewer 2**

*- Line 178 The citation here can be a little misleading, as the EnKF in (Evensen 1994; Burgers et al. 1998) are not usually referred to as the deterministic version of EnKF. I suggest putting the citation (Sakov and Oke 2008) here, and move the citations (Evensen 1994; Burgers et al. 1998) to line 180. For the EnKF reference, in addition to (Evensen 1994; Burgers et al. 1998), I recommend that also include another reference (Houtekamer and Mitchell, 1998)*

Section 3 (Data assimilation scheme, line 169-226) has been restructured, and now includes the suggested citations and makes a clearer difference between the EnKF and DEnKF. In particular, the approximated ensemble transform matrix used in the DEnKF has been specified in the text. Furthermore, the section provides a higher level of detail on the configuration of the EnKF DA scheme.

*- Line 183-184: Although it is tangent to the main thread of the paragraph here, van Leeuwen (2020) notes that in the original stochastic EnKF, the perturbations should be added to the ensemble equivalence of the observation H(x) instead of the observation y. This distinction becomes significant when the observation error is non-symmetric (e.g., skewed), which can have important implications for, e.g., bounded observations. van Leeuwen, PJ. (2020) A consistent interpretation of the stochastic version of the Ensemble Kalman Filter. QJR Meteorol Soc., 146: 2815– 2825. https://doi.org/10.1002/qj.3819*

We have included a remark in Section 3.1 (line 192-195).

*- Line 185-190: Equation (2) is the stochastic version of EnKF, while this paper uses the deterministic version of EnKF. Using Equation (2) here can be confusing to the readers.* **Therefore, I suggest, e.g., replacing Equation (2) with the deterministic transform equation in Sakov and Oke (2008) and replacing this paragraph with a new one (or add a new one) for the deterministic EnKF in Sakov and Oke (2008).**

The revised manuscript contains a separate paragraph (line 188-196) on the deterministic EnKF (DEnKF); in particular, the approximated ensemble transform matrix used in the DEnKF is specified in the text together with the reference to Sakov and Oke (2008). Prior to this paragraph, an introductory paragraph about the EnKF is kept with the corresponding general EnKF equations (Eqs. 2 and 3) and references. The suggested reference of Houtekamer and Mitchell (1998) was added as an EnKF reference in the text (line 170-171).

*- Line 197-198 Is this a reasonable assumption for the observations assimilated in this work? This assumption will introduce larger representation error to the observations that are taken at time points more distant from the analysis time. Although this issue is discussed in Section 6.4, I suggest adding one or two sentences commenting on this assumption here.*

Synchronous assimilation of observed variables is clearly a simplification, and currently we are working on experiments to study the benefits of asynchronous assimilation which takes into account the observation time in the analysis. We have added a discussion on the implications of this assumption in Section 8 (Section 8.2, line 589-592).

*"The current DA setup in Barents-2.5 evaluates all observations at analysis time 00 UTC. For slowly varying model state parameters, such as the hydrography at depth, this assumption bears limited consequences as long as the model error is larger than that of the diurnal variability of the observed parameter. Ongoing development work on Barents-2.5*

*focuses on assimilation of SIC and SST observations at the observation time, showing positive improvements for modeling SIC."*

We are currently working on an analysis of the benefits of assimilating sea ice concentration at observation time (asynchronous assimilation), which shows improvement of model results. A manuscript on this work is in preparation, and we have planned to undertake a similar study on assimilation of SST at observation time. This is beyond the scope of this paper, but we included a footnote commenting on the ongoing work. A discussion on future perspectives towards asynchronous assimilation is added in Section 8 (Section 8.4, line 616-619).

*"Asynchronous assimilation of SIC, i.e. swath data, is in a development stage and SST assimilation will also likely benefit from asynchronous assimilation. In asynchronous DA, each satellite swath is compared to the model field at the respective retrieval time instead of applying observations at a fixed time of the day. This may lead to a reduction of the representation error in observations and hence introduce a tighter constraint on the model state."*

*- Line 199-202 I suggest extending this paragraph by adding more DA details. Specifically, (1) Including some details about what the "spread reduction factor" and the "global moderation factor" mean, and how they work.*

The parameters used to configure the EnKF are given in Tab. 4 and we have added explanations about the configuration parameters for the EnKF in Section 3.3, i.e. the localisation radius, moderation factor, and exaggeration parameter (wrongly called spread reduction factor in the previous manuscript version). More insight on inflation and localisation is also provided in the discussion in Section 8.2 (line 563-570).

*(2) It seems that only horizontal localization is applied. Do the sea surface observations have impact on the state variables in the ocean (e.g., the ocean current at 30-m deep)?*

It is true that only horizontal localisation is applied for the EnKF in our model, and we clarify this in Section 3.3 (line 223-226). With further investigation of model results following this remark, we note in fact that in some circumstances the lack of vertical localisation combined with relatively low number of ensemble members can lead to artifacts in the model analysis at depth, possibly resulting from spurious correlations in the ensemble. We now discuss this point in Section 8.2:

*"The EnKF DA scheme relies on a statistical description of possible model states compared to the available observations, and therefore the ensemble needs to cover the degrees of freedom for the ocean state within a localisation radius (Tab. 4), which is maintained in the current setup using 24 ensembles. Only horizontal localisation is applied here, but we note that the system could benefit from vertical localisation as well."*

*I suggest providing more information on how the observation errors are moderated, since it is one of the key part in DA system. I wonder if the adaptive tuning of the observation error could somehow partially compensate the issue of the time-dependent representation error for the observations.*

A brief description of the role of observation error moderation is now given in Section 3.3 (line 212-222):

*"The EnKF-C software package allows configuration of the DA scheme in terms of an exaggeration parameter R, a localisation radius $r_{loc}$ for each observation type (Tab. 4), a global moderation factor K = 1.5, and an inflation factor. ... K regulates the moderation of the observation impact by smoothly increasing the observation error as a function of the innovation magnitude. For large innovations, K plays an important role in the increase of the observation error; for small innovations, observations errors are mostly kept unchanged. For details on the observation error moderation we refer to Sakov (2014)."*

*Are the observations the same type of variables as the model states in ROMS/CICE? i.e., are the observation operators just interpolating the model state to the observation location?*

At present, our model system directly maps the model state variables to observed parameters, which we have clarified in Section 3.3 (line 203-205) and Section 8.1 (line 581-582). We think that this is a fair assumption for in-situ hydrography and sea ice concentration from passive microwave imagery. For sea surface temperature as observed by infrared and visible band imagery, we acknowledge a mismatch during skin temperature and upper model layer temperature. Due to the far-north location of the model domain, this mismatch is limited but visible during spring and summer time. We are currently working on the implementation of an observation operator for the skin temperature. During most of the time however, our model domain is either well-mixed due to wind forcing or experiencing relatively low levels of radiative forcing that drive the differences between upper layer and skin temperatures. This discussion is addressed in Section 8.2 of the revised manuscript (line 581-588).

*- Line 291 Why are the SST validated for these regions (defined in Fig. 6) separately? Are there any important implications from Fig. 7? I suggest adding one or two sentences briefly discussing the results from Fig. 7.*

Fig. 5 (previously Fig. 6) has been modified slightly to better distinguish the regions for validation (numbered from 1 to 12), and also explicitly state the regions in the figure caption. Each area's distinct hydrography is the reason for validating these regions separately, which is now explicitly stated, and we also mention that the Arctic CMEMS product is validated for the same areas (line 337-339).

*- Line 304 What kind of failures in DA are specifically referred to here?*

Most common have been IT-related issues (e.g. lack of sufficient memory allocated to the EnKF task) and lack of exceptions dealt with in the queuing system (e.g. failures of the scheduling system and human mistakes during code updates). At few instances, the EnKF-c software failed due to observations or model states that resulted in the software exiting. We have specified this in the revised manuscript (line 321-326), however using a low level of detail because it is not central in this paper and very specific about the local infrastructure.

*- Line 347-349 I suggest including the reference(s) for the rank histogram, e.g., (Hamill 2001). Hamill, T.M. (2001) Interpretation of rank histograms for verifying ensemble forecasts.*

Thank you for suggesting this reference, it has been included in line 349.

*- Lines 353-355: I suggest incorporating more descriptions on how the reliability diagram is generated, and the way to interpret the reliability diagram.*

In the new method section, Section 6 Validation methods, we have provided a description of the method and how to interpret the resulting diagram (line 367-389). The resulting diagram (Fig. 14) is discussed in 7.4 Ensemble spread (line 465-473), and a specific description is presented in detail in the figure caption.

*- Line 366-367: Similar to the previous comment, e.g., why does the reversed S-shape indicate low ensemble spread?- Line 374-396 (general comment for section 5.4)*

We have provided a description of the methods for assessing ensemble spread in Section 6 of the revised manuscript (line 379-385) and have added a reference (Bröcker and Smith 2007).

*It is interesting to also discuss the analysis increment for the unobserved variables, e.g., ocean current.*

In the revised manuscript, we have briefly addressed the increments for increments of surface current and sea level height (line 503-506).

*"Model state variables that are not observed are adjusted by the EnKF along the observed variables, based on the covariance matrices between each variable. This includes the 3D current field, surface elevation, and all ice state variables. Surface current increments are at the order of 0.05 m/s and sea surface elevation increments are at the order of 0.01 m and mostly associated with alterations in the density field due to modifications in temperature fields."*

*- Line 384 I suggest being more specific here. For example, revise "The correlation for SST is..." to "the correlation between ... and ... is"*

We have rephrased this part (line 487-488) to be more specific, and it now reads: "*The correlation between modeled and observed SST is slightly improved [...].*"

*- Line 481-483 While I do not insist on conducting more DA experiments to address the following issues in this paper, it would be helpful to include some of the discussions regarding the following questions: (1) Could a larger inflation factor lead to a better DA and forecast performance? (2) Exploring whether techniques to address insufficient ensemble spread, e.g., see the list below, can be effective would be an interesting experiment in the future work as well.*

During DA experiments, we have seen that larger inflation factors for sea ice variables do improve the validation metrics for sea ice concentration. However, as a result of the inflation we also get artifacts in other variables such as sea ice thickness and current velocities. We therefore keep inflation to relatively low levels. Section 3.3 (line 214-216) is more specific:

*"This inflation factor is set to 1.05 (5% inflation) for all model variables except for SIC using 1.1 (10% inflation); a higher inflation factor is required for SIC compared to other variables because of its lower ensemble spread, which is particularly needed around the ice edge area."*

We find great interest in the methods provided by the reviewer and mention these as possible ways forward in the revised paper, along with using perturbation techniques to increase ensemble spread. Line 564-570 now reads:

*"The ensemble spread is modified through i) reduction by EnKF analysis step, ii) increase during the model integration, and iii) inflation of the analysis increments during the DA*

*analysis. The inflation allows us to control the ensemble spread and avoid collapse. A moderate inflation factor is used for most variables, and slightly higher for SIC which suffers low spread. While higher inflation has the potential to further improve the spread in SIC, we also experience that large inflation factors lead to multivariate artifacts in the model analysis. Anderson (2009) propose a spatially and temporally varying adaptive inflation algorithm to allow for a larger covariance inflation, leading to a more efficient increase of ensemble spread. El Gharamti (2018) iterates on this method, improving the stability of the adaptive inflation to the occurrence of negative and physically intolerable inflations."*

We also return to these plans in the outlook section (line 621-626) of the revised manuscript, but are not yet ready to provide results.

*(3) Since SIC is a bounded variable, assuming Gaussian error for SIC is inappropriate especially when SIC value is close to the boundary (i.e., zero or one). This non- Gaussianity can make Gaussian DA method, like EnKF, sub-optimal. It would be helpful to check the ensemble distribution of SIC at a single location before and after DA in a single DA cycle, (1) when SIC observation is close to the boundary, e.g., SIC = 0 or 1, (2) when SIC observation is away from the boundary, e.g., SIC = 0.5. Using some non-Gaussian DA techniques (e.g., Bishop 2016; Poterjoy 2016; Hu and van Leeuwen 2021; Anderson 2022; Chan et al. 2023, etc) can alleviate this problem and may improve the assimilation and the forecast of SIC.*

We followed up on your suggestion and have included a brief analysis of the local distribution of sea ice concentration in the ensemble, for the background and analysis. The distribution at four random points sampled from four different sea ice concentration ranges (10%, 40%, 70%, 90%) are shown in a new figure, Fig. 17. The assumption of gaussianity is addressed and we have included a remark on methods to deal with non-Gaussian variables (and suggested references) in lines 593-599.

–

Finally, we would like to thank both reviewers for their suggestions to improve the manuscript and for their comments about future model development that may improve the forecast capabilities of Barents-2.5 in later model setups. In particular, we look forward to following up with tests using the EAP sea ice rheology in CICE and testing adoptive inflation algorithms.

---

## Referee Report (RR1)

Review for "Barents-2.5km v2.0: An operational data-assimilative coupled ocean and sea ice ensemble prediction model for the Barents Sea and Svalbard"

**Recommendations: Accept for publication**

I highly appreciate the authors' effort to address the issues raised in the previous round of review. The revised version has improved accessibility, and I can better appreciate the work compared to the previous version. Please find below for a few very minor (mostly typos) comments.

**Specific line-by-line comments**
- Line 209:
  … is estimated by the "variance" of the ensemble of background states -> covariance.

  It is good to also mention that the estimate is flow-dependent, which is a key advantage of ensemble DA methods.

- Line 224:
  The reference is off.

- Lines 231-232:
  Is SIC observation linearly related to the model variables (is it the weighted average of several model variables)?

- Line 233:
  "ENKF" -> EnKF

- Line 249:
  The reference is off.

- Lines 417-420:
  It might be helpful to mention that $\bar{P}_b$ and $F_b$ are the x and y axis in Figure 14 respectively.

- Line 431:
  "0>SIC<0.1" => "0<SIC<0.1"

- Lines 545-555:

  "0>SIC<0.4" => "0<SIC<0.4"

- Captions in Figure 8 & 10:

  Why are the RMSEs negative? Should they be ME instead?

- Figure 17:

  I can understand that the post-processed SIC is between 0 and 1, so when the analysis ensemble is outside [0,1] they will be corrected. However, some of the subplots seem to suggest the analysis ensemble members with SIC within [0,1] are also changed. Why?

---

## Author Response (AR2)

*Reply to corrections requested by the Reviewer:*

*Specific line-by-line comments*

*- Line 209: ... is estimated by the "variance" of the ensemble of background states -> covariance. It is good to also mention that the estimate is flow-dependent, which is a key advantage of ensemble DA methods.*

We have corrected this point, and added the remark about flow-dependency into the manuscript.

*- Line 224: The reference is off.*

We removed the reference, and instead refer to the user manual in a footnote.

*- Lines 231-232: Is SIC observation linearly related to the model variables (is it the weighted average of several model variables)?*

Observed SIC is compared to the sum of SIC classes in the model.

*- Line 233: "ENKF" -> EnKF*

corrected

*- Line 249: The reference is off.*

Same as above

*- Lines 417-420: It might be helpful to mention that $\overline{P}$ b and F b are the x and y axis in Figure 14 respectively.*

Even though Pb is related to the quantity on the x axis, they are not exactly the same. Rather, the forecast probabilities show an empirical representation of Pb.

- *Line 431:*

*"0>SIC<0.1" => "0<SIC<0.1"- Lines 545-555:*

*"0>SIC<0.4" => "0<SIC<0.4"*

Corrected

*- Captions in Figure 8 & 10:*

Why are the RMSEs negative? Should they be ME instead?

This was a mistake in the figure caption, we have corrected this in the revised manuscript. Thank you for noting this!

*- Figure 17: I can understand that the post-processed SIC is between 0 and 1, so when the analysis ensemble is outside [0,1] they will be corrected. However, some of the subplots seem to suggest the analysis ensemble members with SIC within [0,1] are also changed. Why?*

The post-processed SIC differs from the EnKF analysis if any of the SIC classes contain values outside the range of $\,<\,SIC\,<\,100\%$. The correction is applied to each SIC separately, and therefore even tolerable values of total SIC could be modified.

*Answers to corrections requested by the Editor:*

*1. The section of methodology should be improved. Particularly, the description of Deterministic EnKF should be better organized. Although the DA system is using the package of enkf-c. The meaning of different parameters should be clearly explained.*

*The deterministic version of the EnKF (Sakov and Oke, 2008, SO08) should be briefly explained. " In the EnKF, …. Based on the assumption of a smack KH term". I don't think this explains how SO08's deterministic EnKF works.*

*The added paragraph does not explain how K=1.5 works?*

*Please clarify why only the horizontal localization is used in this study? (P8 Line 223, 579)*

We have expanded the section describing the DEnKF, and include further references. We also clarify that only horizontal localisation is implemented in the EnKF-C software package.

*2. Although it is good that the authors tried to explain the evaluation metrics in details, the explanations should be concise and can be referred to proper references, such as (Wilks, D.).*

*The reliability diagram is often used for the verification of meteorological ensemble forecast. The paragraph for explaining the reliability diagram can be shortened and referred to the proper reference.*

We have shorted the description of the Reliability diagram, and hope that the present text is concise enough while providing enough explanation. We also add another reference that discuss the contribution of observation errors on rank histogram and reliability diagrams (Saetra et al. 2004).

*3. It should be noted that the designed ensemble forecast in this study does not represent "model uncertainty" since the ensemble members in this study only have differences in the initial states. The model uncertainty is usually referred to the ensemble with perturbed model physics. Please avoid to use "model uncertainty".*

We'd like to thank you for making us aware of this difference, and clearly we do not address the uncertainty in the model physics. Therefore we decided to instead refer to "uncertainty of the forecast model".

*4. The discussion regarding the non-Gaussianity of SIC should be more organized. Also, why is the small ensemble spread of SIC attributed to non-Gaussianity? The small ensemble spread may simply because of the model bias.*

*On P19, Line 556: there is no direct argument to indicate that the small spread of SIC is related to non-Gaussianity. The issue of non-Gaussian SIC was mentioned later at Line 593-599. The discussions should be re-organized.*

We agree that this discussion was scattered across section 8.2. We have pulled these parts together, and thereby also shortened this discussion a bit. We agree that the lack of sufficient ensemble spread may rather be due to a lack of ensemble spread, and provide this explanation revised text.

*5. There are several unclear sentences in the revised manuscript. I suggest the authors should make the clarification.*

*There are many acronyms that the readers may not be familiars with. Please provide a list to give the definition of the **acronyms**, such as TOPAZ, OSTIA…etc.*

The definitions for OSTIA, CTD, HYCOM and MET are now included.  Instead of TOPAZ, we write *Topaz* in italic to indicate that this is a name because the original full name is very long and irrelevant for this paper.

*P8, Line 239: I suggest to remove "some".*

We accept this change.

*Section 4.3: Is there any QC procedure for SIC?*

Not for the observed SIC.

*P11, Line 325: Since a failure of the EnKF update is more common to occur, please clarify "the poor performing statistic representations in the EnKF algorithm". Is this a too small ensemble spread or any kind of situation lead to filter divergence?*

Insufficient ensemble spread is one possible reason, but possibly also the model bias. We include this remark in the revised manuscript.

*P14, Line 410: the availability of the "observations"?*

This refers to the availability of the DA altogether. The text has been clarified.

*P18, Line 506: The argument that "the model analysis and forecast is thought to contain skill if the model error grows with lead time" is not reasonable. Even the forecast error keeps growing, the forecast can be not be regarded as skillful compared to the reference.*

We clearly agree with this remark, and modify this definition:
"The model analysis is skillful compared to the forecast if the model error grows with lead time."

*It is unclear why there is another forecast by combining the model trend with the past observation. What is the purpose?*

The model trend provides a forecast without an initial error from the model, focusing on the model's ability to predict changes in SST. For example, we often see that SST changes are predicted well even if the absolute value is off. We provide this explanation in the validation method part of the revised manuscript.

*P16, Line 469: why does the small ensemble spread partly owes to the observation error? Do the authors suggest to inflate the observation error? The small ensemble spread may also be related to the model bias. Please give a brief explanation. Also, rescaling the forecast probability to match the observed occurrence frequencies is also a solution to remove the impact of model bias on forecast probability (Chang et al. 2015, MWR).*

Not the model ensemble spread, but the presentation in the rank histogram and reliability diagram is affected by the observation error. We include a reference in the revised manuscript (Saetra et al. 2004).

*P19, Line 567-569: Do the authors expect that the adaptive covariance inflation method can mitigate the issue of non-Gaussianity?*

We have not yet conducted such experiments, but see a potential in this method. We do not believe that this method will primarily mitigate the issue of non-Gaussianity, but hope that it can help to mitigate un-physical solutions during the inflation.

*P21, Line 624: "transform observation operators" is not clear. Did the authors mean "transform the observation variable from a skewed to a quasi-Gaussian distribution?*

We have re-written this statement as "In addition, applying transform  operators for SIC, e.g. through remapping of SIC to a Gaussian variable, can mitigate difficulties of the EnKF to assimilate SIC directly…"

In principle, both skewed but also bounded interval distribution could be addressed in such methods. Since we have not undertaken experiments using such methods, it is too early for us to go into more detail about the potential benefits.

Finally, we would like to thank the Editor for their thorough corrections and the handling of our submission to GMD.

[revised manuscript text omitted]